# Novel Thienopyrimidine-Hydrazinyl Compounds Induce DRP1-Mediated Non-Apoptotic Cell Death in Triple-Negative Breast Cancer Cells

**DOI:** 10.3390/cancers16152621

**Published:** 2024-07-23

**Authors:** Saloni Malla, Angelique Nyinawabera, Rabin Neupane, Rajiv Pathak, Donghyun Lee, Mariam Abou-Dahech, Shikha Kumari, Suman Sinha, Yuan Tang, Aniruddha Ray, Charles R. Ashby, Mary Qu Yang, R. Jayachandra Babu, Amit K. Tiwari

**Affiliations:** 1Department of Pharmacology and Experimental Therapeutics, College of Pharmacy and Pharmaceutical Sciences, University of Toledo, Toledo, OH 43614, USA; saloni.malla@rockets.utoledo.edu (S.M.); angelique.nyinawabera@rockets.utoledo.edu (A.N.); rabin.neupane@rockets.utoledo.edu (R.N.); donghyun.lee@rockets.utoledo.edu (D.L.); mariam.aboudahech@rockets.utoledo.edu (M.A.-D.); shikha.kumari@rockets.utoledo.edu (S.K.); 2Department of Pharmaceutical Sciences, College of Pharmacy, University of Arkansas for Medical Sciences, Little Rock, AR 72205, USA; rpathak2@uams.edu; 3Institute of Pharmaceutical Research, GLA University, Mathura 281406, UP, India; suman.sinha@gla.ac.in; 4Department of Bioengineering, College of Engineering, University of Toledo, Toledo, OH 43606, USA; yuan.tang@utoledo.edu; 5Department of Physics, College of Math’s and Natural Sciences, University of Toledo, Toledo, OH 43606, USA; aniruddha.ray@utoledo.edu; 6Department of Pharmaceutical Sciences, College of Pharmacy, St. John’s University, Queens, NY 11439, USA; cnsratdoc@optonline.net; 7MidSouth Bioinformatics Center and Joint Bioinformatics Graduate Program of University of Arkansas at Little Rock, University of Arkansas for Medical Sciences, Little Rock, AR 72204, USA; mqyang@ualr.edu; 8Department of Drug Discovery & Development, Harrison School of Pharmacy, Auburn University, Auburn, AL 36849, USA; ramapjb@auburn.edu

**Keywords:** triple-negative breast cancer, multidrug resistance, non-apoptotic cell death, thienopyrimidines, mitochondrial membrane potential, dynamin-related protein 1 (DRP1)

## Abstract

**Simple Summary:**

Triple-negative breast cancer (TNBC) is characterized by the absence of estrogen receptors, progesterone receptors, and human epidermal receptors. This lack of receptors renders TNBC unsuitable for targeted-based treatment, making it the most fatal and aggressive subtype of breast cancer. TNBC has a greater relapse rate, worse prognosis, and increased metastasis rate compared to non-TNBC because of its tendency to resist apoptosis, a programmed cell death triggered by most chemotherapeutic drugs, producing anticancer efficacy. This work describes two new drugs, TPH104c, and TPH104m, that induce a non-apoptotic form of cell death in TNBC. The incubation of TNBC cells with TPH104c or TPH104m causes cellular expansion and rupture without producing apoptotic characteristics, such as nuclear fragmentation, apoptotic blebbing, or caspase activation. TPH104c and TPH104m decreased the mitochondrial protein, division regulator, and dynamin-related protein 1 (DRP1). The level of DRP1 in TNBC cells affects the magnitude of cytotoxicity produced by TPH104c and TPH104m.

**Abstract:**

Apoptosis induction with taxanes or anthracyclines is the primary therapy for TNBC. Cancer cells can develop resistance to anticancer drugs, causing them to recur and metastasize. Therefore, non-apoptotic cell death inducers could be a potential treatment to circumvent apoptotic drug resistance. In this study, we discovered two novel compounds, TPH104c and TPH104m, which induced non-apoptotic cell death in TNBC cells. These lead compounds were 15- to 30-fold more selective in TNBC cell lines and significantly decreased the proliferation of TNBC cells compared to that of normal mammary epithelial cell lines. TPH104c and TPH104m induced a unique type of non-apoptotic cell death, characterized by the absence of cellular shrinkage and the absence of nuclear fragmentation and apoptotic blebs. Although TPH104c and TPH104m induced the loss of the mitochondrial membrane potential, TPH104c- and TPH104m-induced cell death did not increase the levels of cytochrome c and intracellular reactive oxygen species (ROS) and caspase activation, and cell death was not rescued by incubating cells with the pan-caspase inhibitor, carbobenzoxy-valyl-alanyl-aspartyl-[O-methyl]-fluoromethylketone (Z-VAD-FMK). Furthermore, TPH104c and TPH104m significantly downregulated the expression of the mitochondrial fission protein, DRP1, and their levels determined their cytotoxic efficacy. Overall, TPH104c and TPH104m induced non-apoptotic cell death, and further determination of their cell death mechanisms will aid in the development of new potent and efficacious anticancer drugs to treat TNBC.

## 1. Introduction

Triple-negative breast cancer (TNBC) is a subtype of breast cancer that does not express estrogen, progesterone, and HER2 receptors and accounts for 15–20% of breast cancer cases [1]. TNBC exhibits a wide range of morphological, genetic, and clinical variations, and has significant aggressive characteristics [2,3]. TNBC patients have a worse survival rate than non-TNBC patients due to a poorer prognosis and a greater recurrence rate [4,5]. Furthermore, TNBC cannot be treated with hormonal therapy, such as selective estrogen receptor modulators (SERMS), HER2 antagonists, or aromatase inhibitors, which limits treatment options [6]. Currently, there are only limited combinations of immunotherapy and chemotherapy for the treatment of metastatic TNBC (mTNBC). These include atezolizumab (Tecentriq) in combination with nab-paclitaxel and pembrozulizumab (Keytruda) and in combination with paclitaxel-carboplatin, doxorubicin-cyclophosphamide, or epirubicin-cyclophosphamide [7,8,9,10]. Similarly, sacituzumab govitecan (Trodelvy) has been approved for patients previously treated for mTNBC with two targeted therapies, and talazoparib (Talzenna) and olaparib (Lynparza) have been approved for mTNBC that contain germline BRCA mutations [11]. Nevertheless, over the past three decades, neoadjuvant anthracyclines, such as doxorubicin and epirubicin, and taxane-based chemotherapeutic regimens, such as paclitaxel and docetaxel, are still the main therapeutic options for patients with early stage TNBC and higher mortality risk [12,13,14]. Although TNBC patients initially have a therapeutic response to chemotherapy, treatment eventually becomes ineffective after the tumor metastasizes or becomes resistant to chemotherapy [13]. Therefore, treatment will not be therapeutically optimal due to an increase in chemoresistance as well as the occurrence of severe toxicity [15,16,17]. 

Resistance to anticancer drugs can be produced by various mechanisms, such as (1) increased expression of certain ATP-binding cassette (ABC) and other efflux transporters [18], (2) resistance to or evasion of apoptosis [19], (3) impairment or decrease in anticancer drug uptake into cells [20], (4) increased DNA damage response and repair [21], (5) increased tolerance of cancer cells to a stressful or non-homeostatic tumor microenvironment [22], (6) mutations in drug targets that significantly decrease or abrogate the interaction of drugs with their cellular targets [23], (7) sequestration of drugs by certain cellular organelles, which decreases the amount of drug that can interact with their cellular target(s) [24,25,26], and (8) increased systemic or intracellular metabolism of drugs to less efficacious or inactive metabolites [27]. Currently, there is an urgent demand for novel anticancer therapies that can circumvent resistance in cancer cells. 

Recently, there has been an increase in the discovery and development of anticancer treatments that induce cancer cell death by mechanisms independent of apoptosis, also known as non-apoptotic cell death, which can surmount resistance to drugs that produce their efficacy by inducing apoptosis [28,29,30]. Mitochondrial dynamics have garnered attention as a potential target for the treatment of cancer [31,32,33,34]. Mitochondrial fission and fusion events, the main processes in mitochondrial dynamics, are involved in maintaining cellular homeostasis [35]. Through this process, cells regulate the number, location, and shape of their mitochondria to meet their energy demands [36]. Dynamin—related protein 1 (DRP1) belongs to the dynamin family of GTPases and regulates mitochondrial dynamics. Upregulation of DRP1, a key protein in mitochondrial fission and a fragmented mitochondrial pattern, occurs in a number of different types of cancer [37,38,39,40]. DRP1 has been implicated in increasing the proliferation, migration, and invasiveness of cancers of different origins [31], including breast cancer [38]. In addition, enhanced mitochondrial fission caused by the upregulation of DRP1 has been correlated with poor prognosis in TNBC [40]. Therefore, the development of a potential pharmacophore to target DRP1 and induce non-apoptotic death could be a milestone in addressing drug resistance related to apoptotic cell death. 

Heterocyclic compounds containing fused thiophene and pyrimidine rings, such as thieno [2,3-d]pyrimidine, possess structural similarities to purine bases and have been reported to be effective as antibiotics [41], antiviral [42], anti-inflammatory [43], antimicrobial [44], anti-tuberculosis [45], antioxidant [46], and anticancer drugs [38,39,40,41,42,43,44]. Furthermore, these compounds can inhibit certain kinases and efflux transporters [47]. Currently, three thieno-pyrimidine-based lead compounds, apitolisib (phosphoinositide-3-kinase (PI3K) and mammalian target of rapamycin (mTOR) kinase inhibitor), SNA-314 Ph1 and pictilisib (inhibitor of PI3Kα and PI3Kδ), are being evaluated for their anticancer efficacy in clinical trials [48,49,50,51,52]. These compounds highlight the importance of thieno-pyrimidine pharmacophores in the development of novel chemotherapeutic molecules. Therefore, in this study, our group has synthesized TPH104c and TPH104m, derived from the parent compound, (*E*)-4-methoxy-2-((2-(5-(*p*-tolyl)thieno [2,3-*d*]pyrimidin-4-yl)hydrazineylidene)methyl)phenol (TPH104) [53,54] and evaluated whether they induce non-apoptotic cell death in TNBC and whether these compounds we found that these compounds dysregulated mitochondrial dynamics by downregulating the protein, DRP1. 

## 2. Materials and Methods

### 2.1. Cell Lines and Cell Culture

All triple-negative breast cancer (TNBC) cell lines (MDA-MB-231, MDA-MB-468, and BT-20), non-TNBC cell lines (ZR-75-1 and MCF-7), and mouse embryonic fibroblast line (MEF) were generously provided by the late Dr. Gary Kruh (University of Illinois at Chicago, Chicago, IL, USA). The normal human mammary epithelial cell line (HMEC) (Cat: PCS-600- 010), MCF-10A (Cat: CRL10317), and MCF-12A (Cat: CRL10782) were purchased from the American Type Culture Collection (ATCC, Manassas, VA, USA). Likewise, Paclitaxel-resistant SUM159 (PAC200) was developed in collaboration with Dr. Dayanidhi Raman (University of Toledo, Toledo, OH, USA) [55]. TNBC cells were cultured as adherent monolayers in Dulbecco’s modified Eagle medium (DMEM) (Corning, Tewksbury, MA, USA). HMEC and MCF-12A were cultured in Mammary Epithelial Cell Basal Medium (ATCC) and Mammary Epithelial Cell Growth Basal Medium (Lonza, Basel, Switzerland), whereas MCF-10A was cultured in DMEM:F12 medium along with supplements as described in this study [56]. These culture media were supplemented with 10% fetal bovine serum (FBS) (Biotechne, Minneapolis, MN, USA), 1% penicillin and streptomycin (Cytiva, Marlborough, MA, USA), and 0.1% plasmocin (Invivogen, San Diego, CA, USA) in a humidified incubator with 5% CO_2_ at 37 °C. All cells tested negative for fungus and mycoplasma. 

### 2.2. Cell Cytotoxicity Assay

The cytotoxic efficacy of thienopyrimidine derivatives was determined as previously described, using 3 different assays: (1) (3-(4,5-dimethylthiazol-2-yl)-2,5-diphenyltetrazolium bromide) (MTT) (Avantor, Radnor, PA, USA), (2) CellTiter-Blue (CTB) (Promega, Madison, WI, USA), and (3) sulforhodamine B (SRB) (TargetMol, Boston, MA, USA) assay. For these assays, cells were harvested using 0.05% trypsin-ethylenediamine-tetraacetic acid (EDTA) (Corning, Corning, NY, USA), seeded at a density of 3000–5000 cells/well in a 96-well plate, and incubated at 37 °C overnight. The next day, the cells were incubated with different concentrations of the test compounds (0.1, 0.3, 1, 3, 10, 30, or 100 μM) and incubated for 72 h. TPH104c and TPH104m were prepared in dimethylsulfoxide (DMSO) (Avantor, Radnor, PA, USA) and then diluted in media to achieve the desired concentration. In contrast, vehicle control (cells incubated in drug-free medium) contained less than 0.1% of DMSO.

For the MTT assay, 4 mg/mL MTT was added to the cells and incubated for 3 h to allow for the conversion of MTT (yellow) to formazan crystals (dark blue). Following incubation, the media was aspirated and DMSO was added to solubilize the formazan crystals. The Cytation 7 Cell Imaging Multi-Mode Reader (Agilent Technologies, Winooski, VT, USA) was used to measure the absorbance at 570 nm.

For the SRB assay, the cells were fixed with 50% (*w*/*v*) tri-chloroacetic acid (TCA) (Avantor, Radnor, PA, USA) and incubated at 4 °C for 1 h. The cells were washed 4 times with deionized, distilled water, and air-dried at room temperature. The next day, the cells were stained with 0.04% (*w*/*v*) SRB solution for 1 h at room temperature and washed 4 times with 1% (*v*/*v*) acetic acid (Avantor, Radnor, PA, USA). After airdrying the plates overnight, 10 mM Tris base solution (pH 10.5) was added to each well and pipetted thoroughly to dissolve the dye, and absorbance was read at 510 nm.

For the CTB assay, the CTB reagent was added to cells and incubated for 3 h to allow the reduction of resazurin (emits a blue color and low fluorescence) by metabolically active cells to resorufin (emits a pink color and high fluorescence). Finally, the plates were shaken for 10 s in a plate shaker, and fluorescence was measured at 560/590 nm.

### 2.3. Real-Time Cytotoxicity Assays 

#### 2.3.1. IncuCyte^TM^ Live-Cell Morphology Study

Real-time morphological assessment of the cells was performed using the IncuCyte^®^ S3 Live-Cell Analysis System (Essen BioScience, Ann Arbor, MI, USA), as previously described [53]. Briefly, BT-20 TNBC cells were plated at a cell density of 3000/well and incubated overnight. The next day, cells were incubated in media with or without different concentrations of TPH104c and TPH104m. The plate was incubated in the IncuCyte^®^ S3 Live-Cell Analysis System to capture images every 6 h for 72 h. The integrated IncuCyte S3 software version 2020B was used to analyze the images.

#### 2.3.2. IncuCyte^TM^ Cytotox Green Assay

The real-time assessment of dead BT-20 cells was conducted using the IncuCyte cytotox green reagent (Essence BioScience, Ann Arbor, MI, USA), as described previously [57]. This reagent enters dead or non-viable cells, due to structurally compromised cell membranes and binds to nuclear DNA, resulting in green fluorescence [58]. BT-20 cells were cultured at a density of 3000 cells/well and incubated overnight. The cells were incubated with different concentrations of the test compounds or vehicle, prepared at a 3X concentration in cytotox dye—containing media. The plate was placed in an IncuCyte^®^ S3 Live-Cell Analysis System, which was programmed to obtain images of BT-20 cells, every 6 h for 72 h. Finally, the integrated IncuCyte S3 software version 2020B was used to analyze the mean fluorescence intensity of the cytotox dye in the BT-20 cells.

### 2.4. β-Galactosidase Staining

BT-20 cells were plated at 4000/well into a 96-well plate and incubated overnight. The next day, the cells were incubated with 2 or 5 μM of TPH104c and TPH104m. Four thousand cells/well of senescent MEFs were also seeded. The plate was incubated for 24 h. Once 80–100% confluency was obtained, mouse embryonic fibroblast (MEF) cells were incubated with 250 nM of doxorubicin for 24 h to induce senescence (positive control), followed by a change in media. The cells were incubated for another week and the media was changed every 3–4 days. 

The next day, the cells were washed with PBS and fixed with 3% formaldehyde for 5 min. The cells were washed with PBS twice, followed by an addition of β-galactosidase (Research Products International, Racine, WI, USA) stain. The cells were placed in a 37 °C incubator with no CO_2_ overnight and imaged to detect β-galactosidase staining the next day using the Color Brightfield channel on the Cytation 7 Imaging Multi-Mode Reader. 

### 2.5. Colony Formation Assay

BT-20 cells were cultured at a density of 500 cells/well into a 6-well plate overnight. The following day, the cells were incubated with 0.1, 0.3, or 1 μM of TPH104c or TPH104m, as well as media containing no test compounds (vehicle control). The vehicle or test compounds were added every 72 h over a 10-day period. The media was carefully aspirated on the tenth day, and the colonies formed in each plate were fixed using 100% methanol. The colonies were stained with crystal violet dye prepared at 0.1% concentration for 15 min in the dark. Finally, the colonies were visualized, using an EVOS microscope at 4 and 20× (Thermo Fisher Scientific, Wayne, MI, USA), and the area covered by colonies was calculated using ImageJ software Version 1.53k (National Institutes of Health, Bethesda, MD, USA).

### 2.6. Cell Cycle Analysis

Briefly, 250,000 cells/well of BT-20 cells were plated in a 6-well plate and allowed to grow overnight. The next day, the cells were incubated with vehicle, 0.5, 1, or 2 μM of TPH104c or TPH104m. Twenty-four hours later, the cells were washed with DPBS, trypsinized using 0.05% trypsin and 2.21mM EDTA, re-washed one time, and suspended in 1ml of ice-cold PBS. Subsequently, 200 μL of propidium iodide (PI), prepared at a stock concentration of 50 μg/mL, was added to each sample, and each sample was incubated for at least 15 min to stain the cellular DNA. The distribution of BT-20 cells incubated with media, TPH104c or TPH104m, was determined in the Go, S, G1, and G2 phases of the cell cycle using a BD Accuri™ flow cytometer (BD Biosciences, Becton-Dickinson, San Jose, CA, USA). One hundred thousand cellular events were collected, with a maximum rate of 1000 events per second, and the results were generated by analyzing the raw instrument files, using FCS express 7 plus De Novo software (Glendale, CA, USA).

### 2.7. Nuclear Staining

To determine whether our lead compounds induced nuclear fragmentation, nuclear staining was conducted according to a previous study [59], using the dye Hoechst 33342 (2′-(4-Ethoxyphenyl)-5-(4-methyl-1-piperazinyl)-2,5′-bi-1H-benzimidazole trihydrochloride) (Immunochemistry Technologies, Davis, CA, USA). Briefly, 100,000 cells/well of BT-20 cells were plated in a 6-well plate containing clear sterile coverslips. Cells were incubated with vehicle or TPH104c (2 or 5 μM), TPH104m (2 or 5 μM), or paclitaxel (1 μM; an anticancer drug that induces fragmentation of cancer cells) [60,61], for 24 h. Next, the cells were stained with 0.5% *v*/*v* of the Hoechst 33342 dye for 15 min. 4% paraformaldehyde at a concentration of 4%, was used to fix cells at room temperature for another 15 min. After 1 wash with PBS, the cells were mounted on a clear slide, using Fluoromount-G (SouthernBiotech, Birmingham, AL, USA). The slides were visualized to detect nuclear fragmentation using the Cytation 7 Cell Imaging Multi-Mode Reader (Agilent Technologies, Winooski, VT, USA), which provided data related to nuclear fragmentation. 

### 2.8. Apoptosis and Mitochondrial Membrane Potential

Alexa Flour 488-conjugated Annexin V (Molecular Probes Inc., Invitrogen, Eugene, OR, USA) staining was used to quantitatively assess apoptosis by flow cytometry, according to a previous study [62]. Briefly, 250,000 cells/well of BT-20 cells were plated in a 6-well plate and incubated with vehicle or lead compounds, TPH104c or TPH104m, at several concentrations (0.5, 2.5, or 5 μM), for 24 h, on the following day. Subsequently, the cells were collected, subjected to a cold PBS wash and resuspended in 1X Annexin-binding buffer. Following a 15 min incubation with 5% Annexin V, 400 μL of 1X Annexin-binding buffer was added. The cells were mixed gently and analyzed for Annexin V staining, using a BD FACSCalibur Flow Cytometer. One hundred thousand cellular events were collected, with a maximum rate of 1000 events per second and the raw data were analyzed using FCS express 7 plus De Novo software.

The mitochondrial membrane potential (MMP) was determined using fluorescence microscopy by staining cellular mitochondria with tetramethylrhodamine ethyl ester (TMRE) dye (Invitrogen, Waltham, MA, USA). Briefly, 3000 cells/well of BT-20 cells were seeded in a 96-well plate and incubated with 2 or 5 µM TPH104c or TPH104m, for 24 h. As a positive control, cells were incubated with carbonyl cyanide 3-chlorophenylhydrazone (CCCP) (TargetMol, Boston, MA, USA), a compound that uncouples mitochondrial oxidative phosphorylation [63], at a concentration of 200 μM, for 4 h. After incubation with the test compounds, the media was carefully removed from each well and incubated for 20 min with 0.4 µM of TMRE in PBS. Finally, the dye was carefully removed and replaced with fresh PBS. The cells were then imaged to detect MMP using a Cytation 7 Cell Imaging Multi-Mode Reader.

### 2.9. Cell Lysis and Western Blot Analysis

Western blot assays were conducted to determine the effect of TPH104c and TPH104m on the levels of cleaved and full-length caspases (-3, -7, -8, -9), B-cell lymphoma-2 (BCL-2)—associated X (BAX), BCL-2 homologous antagonist/killer (BAK), BCL-2, cleaved and full-length Poly (ADP-ribose) polymerase (PARP), which are increased during apoptosis [64]. Briefly, cells were seeded and incubated with vehicle, TPH104c, or TPH104m (0.5, 1, 2, or 5 µM) for 12 h. Subsequently, the cells were washed with ice-cold PBS. The subcellular fraction was obtained by scraping the cells using a cell scrapper in ice-cold cytosolic lysis buffer [65]. Next, the lysates were left on ice for 15 min. Nonyl phenoxypolyethoxylethanol (NP-40) (10% *v*/*v*) was introduced in the lysates and subjected to 2–3 min incubation. The lysates were then centrifuged for 10 min at 4 °C and 14,000 rpm. The resultant protein concentration in the lysates was determined using bicinchoninic acid (BCA) assay. Thirty micrograms of the proteins were loaded and separated on a 10% acrylamide SDS-PAGE gel, transferred to a PVDF membrane, and incubated overnight at 4 °C, with primary antibodies against cleaved caspase-3, caspase-3, cleaved caspase-7, caspase-7, cleaved caspase-9, caspase-9, cleaved caspase-8, caspase-8, BAX, BAK, BCL-2, cleaved PARP, PARP, p-DRP1, DRP1, p-MFF, MFF, FIS1, MFN1, MFN2, OPA1, and β-actin. Apart from β-actin, which was prepared at a dilution of 1:2000, all antibodies were prepared at a dilution of 1:1000 in 5% Bovine serum albumin (BSA). Horseradish peroxidase (HRP)-labeled anti-rabbit or anti-mouse secondary antibodies (1:4000) were added the next day for 1.5 h. G:BOX Chemi XX6/XX9, obtained from Syngene (Frederick, MD, USA), was used for protein band detection in the blots. ImageJ software (National Institutes of Health, Bethesda, MD, USA) was used to quantify the protein. Cellular proteins were quantified as a ratio to β-actin and were normalized to the vehicle control. The densitometry readings/intensity ratio of each band, andthe original whole western blot (uncropped blots) showing all the bands with the molecular weight markers, is shown in the Appendix A.

### 2.10. Caspase-3/7 Activity Assay

Caspase-Glo^®^ 3/7 Assay (Promega, Madison, WI, USA) was used to study the caspase 3/7 activity in BT-20 cells incubated with lead compounds. Briefly, 3000 BT-20 TNBC cells/well were plated in an opaque 96-well plate. The cells were incubated the following day with the vehicle, TPH104c (0.3, 1, or 3 μM), or TPH104m (0.3, 1, or 3 μM) for 24 h. Next, a mixture of Caspase-Glo^®^ 3/7 buffer and lyophilized Caspase-Glo^®^ 3/7 substrate was equilibrated to room temperature. The plates were removed from the incubator and allowed to adjust to room temperature for 30 min. Next, 100 μL of the Caspase-Glo^®^ 3/7 reagent was added to each well, and the samples were placed in a shaker for 30 s. After 3 h of incubation at room temperature, luminescence, which indicates cleavage of initiator caspases (caspase-3 and -7), was measured using a microplate reader (Agilent Technologies, Winooski, VT, USA). 

### 2.11. Immunofluorescence Staining and Analysis

Immunofluorescence assays were conducted as previously described [66]. In total, 100,000 BT-20 cells per well were plated on coverslips in a 6-well plate. On the following day, cells were incubated with either vehicle, TPH104c (2 or 5 µM), or TPH104m (2 or 5 µM) for 24 h. After the incubation period, the media was carefully removed, and the cells were fixed using 1 mL of 4% paraformaldehyde for 15 min at room temperature. The cells were rinsed three times with PBS for 5 min. Five hundred microliters of 0.2% Trition-X100 was added to each well for 10 min to permeabilize the cells. The cells were washed three times with PBS for 5 min each. The cells were then blocked with 3% bovine serum albumin (BSA) in PBS, containing 0.1% Tween, for 2 h at room temperature before incubation with either cytochrome c (1:250), DRP1 (1:50), or phosphorylated-DRP1(1:450) monoclonal antibodies (Cell Signaling Technology, Danvers, MA, USA) overnight at 4 °C. The cells were incubated with either anti-mouse Alexa Fluor™ 594 (Invitrogen, Waltham, MA, USA) or anti-rabbit Alexa Fluor™ 488 at room temperature for 1 h. Nuclei were finally stained with Hoechst 33342 (ImmunoChemistry Technologies, Davis, CA, USA) for 10 min. Images were captured using a Cytation 7 Cell Imaging Multi-Mode Reader.

### 2.12. Molecular Docking Studies

The interaction between the Drp1 protein and ligands (TPH104c and TPH104m) was examined through a docking simulation conducted using the AceDock program accessed through the Playmolecule platform (https://www.playmolecule.com/AceDock/). AceDock is a set of protein-ligand docking protocols that run rDock [67] in the backend. Docking software was used to predict the binding mode of a given ligand to a defined binding site in a protein. The protein data bank (http://www.rcsb.org) was used to retrieve the X-ray crystal structure of DRP1 (PDB-ID: 4H1V) [68]. Subsequently, the protein underwent various optimizations, including dehydration, hydrogenation, refinement of loop regions and selection of the binding site based on the natural ligand in 4H1V. Subsequently, TPH104c and TPH104m were docked into the active binding site of the DRP1 protein using the same parameters. Both compounds were observed to occupy the same pocket (Appendix A). The starting point for our simulations was determined by extracting coordinates from previously docked structures. We conducted Molecular Dynamics (MD) simulations using GROMACS software (version 2023.1) [69,70] and applied the CHARMM36m force field [71]. To prepare the systems, we utilized the CHARMM-GUI web server [72,73,74]. We used a cubic box with dimensions of 10.10 nm^3^, which was subsequently filled with TIP3P water molecules [75], and the system’s charge was neutralized with potassium chloride counter ions, at 0.15 mol/liter. The system underwent optimization, using the steepest descent algorithm [76], to reach its lowest energy state. To maintain the positions of both the ligand and protein atoms, we imposed a position restraint of 1000 kJ/mol·nm^2^. The entire system was equilibrated under the NVT ensemble for 1 nanosecond, during which the V-rescale thermostat was used to regulate the temperature at approximately 310 K. After the NVT step, we proceeded to equilibrate the system under the NPT ensemble for an additional 1 nanosecond, ensuring that the system’s pressure stabilized at 1 atm. Subsequently, we conducted two sets of production MD runs for each ligand, utilizing a 3 fs timestep with hydrogen mass repartitioning [77] and the leap-frog integrator. The only difference between these runs was the assignment of the initial velocity seeds, both starting from the well-equilibrated system at 310 °K and a pressure of 1 atm. For calculating long-range electrostatic effects, we used the Particle-mesh Ewald (PME) algorithm [78], and the length of covalent bonds was constrained using the LINCS algorithm [79], known for its computational efficiency, compared to the SHAKE algorithm [80]. We also used the analytical SETTLE algorithm [81] to reset the positions and velocities to satisfy the holonomic constraints on the rigid water model. Finally, 100 nanoseconds of unbiased simulations were run to examine the ligand dynamics and corresponding protein conformations. Upon completion of the simulations, the protein was repositioned at the center of the simulation box, and the periodic boundary conditions were removed from the trajectory.

### 2.13. Surface Plasmon Resonance (SPR) Binding Assay

The binding of TPH compounds to DRP1 protein was validated, using surface plasmon resonance (SPR) (Nicoya Lifesciences, Kitchener, ON, Canada). His-tagged recombinant Drp1 protein was generously gifted by Dr. Blake Hill (Medical College of Wisconsin). The protein was diluted in a solution of immobilization buffer (pH 7.2) that consisted of HEPES (10 mM), NaCl (0.15 M), Tween 20 (0.05%), and then tethered to a high-sensitivity nitrilotriacetic acid (NTA) sensor chip (Nicoya Lifesciences, Kitchener, ON, Canada) at a final surface concentration of 10,000 RU. The TPH analogs were prepared in a two-fold concentration series from 100–6.25 µM in an immobilization buffer, and 150 μL was injected across the chip at a rate of 50 μL/min for 1 min. Finally, the binding constant KD value was calculated, using Tracedrawer software Version 1.9.2 (Tracedrawer, Uppsala, Sweden).

### 2.14. Generation of Partial and Complete DRP1-KO Gene Models

Partial and complete *DRP1* knockout models of PAC200 (paclitaxel-resistant variant of SUM159) were generated using the CRISPR/Cas9 system (Santa Cruz Biotechnology, Dallas, TX, USA). Briefly, 200,000 cells/well of PAC200 cells were plated in a 6-well plate in 3 mL of DMEM media. After 24 h incubation, the mixture of plasmid transfection medium containing plasmid DNA (CRISPR control (sc-418922) and *DRP1* (sc-400459)) and plasmid transfection medium containing transfection reagent was introduced into each well and allowed to incubate for 48 h. The cells were sorted based on the detection of green fluorescent protein (GFP) with a BD FACSAria™ III High-Speed Cell Sorter manufactured by BD Biosciences (Franklin Lakes, NJ, USA). Dulbecco’s phosphate-buffered saline (DPBS), modified without Ca^2+^ and Mg^2+^ ions, was utilized as a sheath fluid, as recommended by BD Biosciences.

The sorted cells were grown in a small Petri dish. After reaching confluency, the cells were placed in a 96-well plate to allow the development of single-cell colonies. Western blotting was performed to confirm the complete and partial knockout of DRP1 in PAC200 cells.

### 2.15. Statistical Analysis

All experiments were performed at least in triplicate. Results are presented as the mean ± standard error of the mean (SEM). GraphPad Prism (San Diego, CA, USA) was used to analyze the data. The data from colony assay, cell cycle assay, ROS assay, Annexin V staining, caspase-3/7 activity assay, western blotting analysis, and comparison of IC_50_ values of control wild-type, partial, and complete DRP KO PAC200 cells were analyzed using a two-way ANOVA, and post hoc comparisons were performed using Dunnett’s test. The analysis of the IC_50_ data for BT-20 cells preincubated with or without z-VAD-FMK was performed using an unpaired, 2-tailed *t*-test. The mitochondrial membrane potential analysis data were analyzed using one-way ANOVA with Dunnett’s post hoc test. The a priori significance level was *p* < 0.05.

## 3. Results

### 3.1. TPH104c and TPH104m Selectively Decreased the Proliferation of Cancer Cell Lines

Using the MTT assay, we determined the antiproliferative efficacy of the two lead compounds, TPH104c and TPH104m, in the (1) TNBC cell lines, BT-20, MDA-MB-231, and MDA-MB-468 and (2) normal human mammary epithelial cell line, HMEC, MCF-10A, and MCF12A (Table 1). The IC_50_ values of TPH104c for BT-20, MDA-MB-231, and MDA-MB-468 TNBC cells were 0.22 ± 0.06 μM, 0.48 ± 0.16 μM, and 0.45 ± 0.17 μM, respectively. The IC50 values of TPH104m were: (1) 0.18 ± 0.03 μM, 0.47 ± 0.15 μM, and 0.27 ± 0.14 μM for BT-20, MDA-MB-231, and MDA-MB-468 cells, respectively.

Neither TPH104c nor TPH104m resulted in a significant decrease in the proliferation of non-cancerous HMEC cells (IC_50_ values for TPH104c and TPH104m were >5 µM). (Figure 1a,b). The TPH compounds were 15-to 30-fold more selective in decreasing in vitro tumor growth, compared to the normal cells, HMEC, MCF-10A, and MCF-12A. We also determined cellular viability using the CellTiter-Blue^®^ (CTB) and Sulforhodamine B (SRB) assays. The IC_50_ values of TPH104c and TPH104m in the CTB assays for the BT-20 cell line were 0.23 ± 0.06 μM and 0.19 ± 0.08 μM, respectively (Table 2). In the SRB assay, the IC_50_ values of TPH104c and TPH104m for the BT-20 cell line were 0.30 ± 0.07 μM and 0.32 ± 0.16 μM, respectively (Table 2). 

BT-20 cancer cells reached their maximum confluence (~95%) after 72 h of incubation with the vehicle (Figure 1c). The incubation of BT-20 cells with 0.1 μM of TPH104c and TPH104m did not produce a marked cytotoxic effect in BT-20 cells. After incubation with TPH104c at a concentration lower than the IC_50_, the number of BT-20 cells increased from 20% to 50% to 95% at 24 h, 48 h, and 72 h, respectively. Similarly, following the incubation of BT-20 cells with TPH104m at a concentration lower than IC_50_, the number of BT-20 cells increased from 20% to 60% to 90% at 24 h, 48 h, and 72 h, respectively. However, at concentrations greater than the IC_50_ value, i.e., 0.3 μM, the level of BT-20 cell confluence was only increased by a smaller percentage, i.e., from 18% to 25% to 40% at 24 h, 48 h, and 72 h, respectively, for TPH104c and from 20% to 30% to 35% at 24 h, 48 h, and 72 h, respectively, for TPH104m. At 1 μM of either TPH104c or TPH104m, the proliferation of BT-20 cells was approximately 20% at 24, 48, and 72 h) (Figure 1c). These results suggest that BT-20 cell proliferation was significantly decreased over time by incubation with either TPH104c or TPH104m at 0.3 and 1 μM (*p* < 0.0001) for 48 and 72 h, compared to the vehicle control.

We also used the Incucyte^TM^ Cytotox Green assay to further validate the effects of TPH104c and TPH104m on TNBC cell viability. In this assay, the highly sensitive cyanine nucleic acid dye, Cytotox green, penetrates and stains dead or non-viable cells due to a compromised cellular membrane, and upon binding to deoxyribose nucleic acid (DNA), it emits green fluorescence [57]. The incubation of BT-20 cells for 72 h with 0.1 μM of either TPH104c or TPH104m did not significantly alter the level of green fluorescence, compared to the vehicle control (Figure 1d). In contrast, after the incubation of BT-20 cells with 1 µM of TPH104c and TPH104m, for 72 h, there was a significant increase in the fluorescence intensity (*p* < 0.0001), compared to the vehicle control (Figure 1c,d). These in vitro results indicated that TPH104c and TPH104m significantly decreased the growth of BT-20 cells and increased the percentage of dead cells, after 72 h of incubation. This was in contrast to the cells incubated with the vehicle control, which continued to grow, multiply, and remained healthy over time, as shown by the low fluorescence intensity of cytotox green dye in the cells.

Similar to the cytotoxicity findings, TPH104c and TPH104m produced a concentration-dependent decrease in BT-20 colony formation (Figure 1e,f). The incubation of BT-20 cells with 0.3 or 1 μM of TPH104c for 10 days, significantly decreased the colony formation area (*p* < 0.05 for 0.3 and 1 μM), compared to the vehicle control. TPH104m also significantly decreased the area of the BT-20 colonies, compared to vehicle control (*p* < 0.05 for 0.3 μM and *p* < 0.1 for 1 μM). TPH104c and TPH104m significantly decreased BT-20 cell division, resulting in smaller, less dense colonies, compared to the control cells, where the cells proliferated rapidly and formed larger colonies (Figure 1e,f).

### 3.2. TPH104c and TPH104m Arrest the Cell Cycle of BT-20 Cells in the S/G2 Phase

It is well known that anticancer drugs can significantly disrupt the cell cycle of cancer cells [82,83]. Therefore, we stained cells with PI to study the effects of TPH104c and TPH104m on the cell cycle. PI stains the DNA and this allows for the determination of the cell distribution in different phases of the cell cycle: G1, S, and G2 phases, using flow cytometry [84]. The vehicle control had a normal cell cycle distribution, where 6.3%, 81.4%, 5.8%, and 5.1% of the cells were in the subG1, G1, S, and G2 phases, respectively (Figure 2a–d).

However, there was a significant decrease in the % of cells in the G1 phase, following incubation with 2 μM of TPH104c, compared to vehicle control (81.4% and 69.15% at 0 and 2 μM, respectively, *p* < 0.001 at 2 μM, Figure 2a,b). BT-20 cells incubated with TPH104c significantly shifted the cell cycle toward the S (15.4% and 21.4% for 1 and 2 μM, respectively, with *p* < 0.01 and *p* < 0.0001 for 1 and 2 μM, respectively) and G2 phases (14.1% and 12.9% for 1 and 2 μM, respectively, with *p* < 0.01 and *p* < 0.05 for 1 and 2 μM, respectively, Figure 2a,b). Similarly, the percentage of cells in the G1 phase, following incubation with 1 and 2 μM of TPH104m, was significantly decreased, compared to the vehicle control (81.4%, 68.4%, and 61.3% at 0, 1, and 2 μM, respectively, *p* < 0.05 at 1 μM and *p* < 0.0001 at 2 μM, Figure 2c,d). BT-20 cells incubated with TPH104m significantly shifted the cell cycle toward the S (18.4% and 22.1% for 1 and 2 μM, respectively, with *p* < 0.05 and *p* < 0.001 for 1 and 2 μM, respectively, Figure 2c,d) and G2 phases (19.7% and 19.5% for 1 and 2 μM, respectively, with *p* < 0.05 for 1 and 2 μM, Figure 2c,d). Overall, these results indicate that BT-20 cells are arrested in the S and G2 stages of the cell cycle after incubation with TPH104c and TPH104m. 

### 3.3. TPH104c and TPH104m-Mediated Cell Death Occurs Independent of Intrinsic and Extrinsic Apoptosis

Our morphological experiments indicated that the incubation of BT-20 cells with 0.1, 0.3, or 1 μM of TPH104c and 0.1, 0.3, or 1 μM of TPH104m did not produce cellular features indicative of apoptosis, such as a decrease in cell size, blebbing of the cytoplasmic membrane, nuclear fragmentation, and apoptotic body formation (Figure 3a and Figure 4a) [62,63]. However, TPH104c and TPH104m produced an increase in the surface area of BT-20 cells (Figure 3a and Figure 4a) that resembled swelling, which ultimately led to cell death by bursting (Appendix A). Alternatively, senescence, defined as a state of permanent cell growth arrest, produces a flattened, enlarged cellular morphology [85,86]. Therefore, to confirm whether TPH104c and TPH104m induced senescence, we performed β-galactosidase staining. Senescent cells had β-galactosidase activity, known as SA-G-gal, a biomarker of senescence [87]. Our positive control, MEF cells incubated with doxorubicin, appeared flatter and much larger than the BT-20 cells and stained blue, which indicated SA-G-gal staining and the presence of senescence. However, there was no β-galactosidase staining in BT-20 cells incubated with TPH compounds. This finding confirmed that TPH104c and TPH104m do not induce senescence in BT-20 cells (Appendix A).

Therefore, to further validate our hypothesis that TPH104c and TPH104m do not induce apoptosis, we incubated BT-20 cells with a fluorophore-labeled Annexin V dye, which binds to phosphatidylserine that is translocated from the inner plasma membrane to the outer membrane during the early stage of apoptosis [88]. BT-20 cells incubated with 0.5, 2.5, or 5 μM of TPH104c or TPH104m did not significantly increase in the percentage of Annexin V positive cells, compared to the vehicle control (Appendix A). Approximately 90% of the vehicle control cells were viable, whereas only 11% of cells that had Annexin V incorporated into their membranes, i.e., they were undergoing apoptosis. Similar to the vehicle group, BT-20 cells incubated with 0.5, 2.5, or 5 μM of TPH104c and TPH104m did not have significant changes in the percentage of PS exposure. BT-20 cells incubated with 0.5, 2.5, or 5 μM of TPH104c, resulted in 12.1%, 13.9%, and 14.8% Annexin V positive cells, whereas BT-20 cells incubated with 0.5, 2.5, or 5 μM of TPH104m resulted in only 14.0%, 15.6% and 15.6% Annexin V positive cells. These results indicate that neither TPH104c nor TPH104m produced a level of apoptosis that was significantly greater than that of the vehicle, i.e., these compounds did not cause cancer cell death by inducing apoptosis. 

We also studied the effect of TPH104c and TPH104m on nuclear fragmentation, another hallmark of apoptosis, where nuclear chromatin condensation begins at the peripheral surface of the nuclear membrane and ultimately produces fragmentation of the nucleus, known as karyorrhexis [89]. BT-20 cells were incubated with Hoechst 33342 dye, which can penetrate into live or viable cells, where it binds to DNA in adenine-thymine regions and produces a measurable blue fluorescence when exposed to light at 460–490 nm [90]. The incubation of BT-20 cells with 2 or 5 μM of either TPH104c or TPH104m for 24 h did not significantly alter the level of blue fluorescence or the shape of the nucleus, compared to the vehicle control (Figure 3b and Figure 4b). However, BT-20 cells incubated with the positive control, 1 μM of paclitaxel, which has been previously reported to produce nuclear fragmentation [91], significantly increased the level of nuclear fragmentation (Figure 3b and Figure 4b). Thus, these results suggest that at the concentrations and incubation times used in this study, TPH104c and TPH104m did not induce nuclear fragmentation, a process that occurs in the later stage of apoptosis.

We also determined the effect of the TPH compounds on the levels of key regulators of apoptosis, including initiator and executioner caspases, caspase-8, caspase-9, caspase-3, and caspase-7, pro-apoptotic proteins, BAK and BAX, anti-apoptotic protein, Bcl-2, and PARP, using western blot assay. BT-20 cells were also incubated with 1 μM of paclitaxel, which alters the levels of certain apoptotic proteins [85]. BT-20 cells incubated with paclitaxel, significantly upregulated the levels of cleaved caspase-3 (*p* < 0.001), caspase-7 (*p* <0.01), caspase-8 (*p* < 0.0001), caspase-9 (*p* < 0.01),) and total caspase-8 (***, *p* <0.001), compared to the vehicle control (Figure 3c,d and Figure 4c,d). In contrast, the incubation of BT-20 cells with 0.5, 1, 2, or 5 μM of TPH104c or TPH104m did not significantly alter the level of cleaved and total caspases (both initiator and executioner caspases), compared to the vehicle control (Figure 3c,d and Figure 4c,d). Since TPH104c and TPH104m did not induce the cleavage of caspase-3 and caspase-7, it is unlikely that they activated the intrinsic or mitochondrial pathway of apoptosis. Furthermore, there was no caspase-8 cleavage in cells incubated with TPH104c and TPH104m, indicating that TPH104c- and TPH104m-induced cell death is not mediated through the extrinsic apoptotic pathway, as activation of initiator caspase-8 is required to cleave and activate caspase-3 and caspase-7 to induce extrinsic apoptosis [92]. Also, the incubation of BT-20 cells with TPH104c or TPH104m did not significantly alter the levels of (1) BAK; (2) BAX, (3) cleaved PARP, and (4) total PARP, compared to the vehicle control. There was no significant change in the levels of Bcl-2 in BT-20 cells incubated with TPH104c and TPH104m (0.5, 1, 2, or 5 μM), compared to the vehicle control. In contrast, there was a significant decrease in Bcl-2 levels in BT-20 cells incubated with paclitaxel, compared to vehicle control cells.

We used the Caspase-Glo^®^ 3/7 assay, which involves incubating cells with the caspase-3/7 substrate, Z-DEVD-aminoluciferin, a substrate for luciferase that is cleaved by active caspase-3 and caspase-7 [93], to further validate the above results, indicating that the TPH compounds do not induce apoptosis. The cleavage of Z-DEVD-aminoluciferin by caspase-3 or caspase-7, releases aminoluciferin, a substrate for luciferase, which produces luminescence [94]. The incubation of BT-20 cells with 0.3, 1, or 3 μM of TPH104c or TPH104m for 24 h did not significantly induce the activation of caspase 3/7, compared to the vehicle control (Figure 3e and Figure 4e). However, BT-20 cells treated with 0.1 μM of paclitaxel, a compound that activates the intrinsic apoptotic pathway [95,96], significantly increased the level of bioluminescence, indicating activation of caspase-3 and caspase-7 (Figure 3e and Figure 4e). We conducted an additional experiment to show that TPH104c and TPH104m do not induce BT-20 cell death by apoptosis. The pre-incubation of BT-20 cells with 100 μM of benzyloxycarbonyl-valine-alanine-aspartate-FMK (Z-VAD-FMK), an irreversible pan-caspase inhibitor [68], did not prevent cell death after incubation with 0.1, 0.3, or 1 μM of TPH104c or TPH104m, compared to vehicle control (Appendix A). There was no significant difference between the IC_50_ values of BT-20 cells treated with Z-VAD-FMK and BT-20 cells treated with either TPH104c or TPH104m (Figure 3f and Figure 4f). These results suggested that neither TPH104c nor TPH104m induced cell death by activating caspase-3 and caspase-7, i.e., they did not induce apoptosis by the intrinsic pathway. It is important to note that although the incubation of BT-20 cells with TPH104c and TPH104m resulted in a significant loss of the mitochondrial membrane potential, it is possible that the level of cytochrome c release in the cytoplasm was not sufficient to activate caspases and cause apoptotic cell death.

### 3.4. TPH104c and TPH104m-Mediated Cell Death Induced the Loss of the Mitochondrial Membrane Potential, Independent of Cytochrome c Release and Reactive Oxygen Species (ROS) Production

As previously discussed, the majority of clinically used anticancer drugs induce the apoptosis of cancer cells [97]. Apoptosis can be activated through the intrinsic/mitochondrial or extrinsic/death receptor pathways [64,98]. Intrinsic apoptosis is mediated by the apoptotic regulator family known as the B-cell lymphoma-2 (BCL-2) family. This family includes pro-apoptotic proteins such as BAX, BAK, and BCL-2 related ovarian killer (BOK), as well as BH3-only proteins like BCL-2 associated agonist of cell death (BAD), BH3 interacting domain death agonist (BID), BCL-2 interacting killer (BIK), BCL-2 modifying factor (BMF), BCL-2-like 11 (BIM), activator of apoptosis hara-kiri (HRK), NOVA, p53 upregulated modulator of apoptosis (PUMA), and SOUL. Additionally, there are anti-apoptotic survival proteins in this family, such as BCL-2, BCL-extra-large (BCL-X_L)_, BCL-2-like protein (BCL2-L-2) or BCL-W, B-cell lymphoma 2 (BCL-B), BCL-2-related protein A1 (BCL-2-A1), and myeloid cell leukemia-1 (MCL-1) [99]. Numerous studies indicate that intrinsic stimuli, such as cellular stress, DNA damage, excessive levels of reactive oxygen species (ROS), and pro-apoptotic proteins, such as BAK and BAX, are activated either transcriptionally or post-transcriptionally and are translocated to the outer mitochondrial membrane [64]. Due to pore formation on the outer surface of the mitochondria, apoptogenic factors such as cytochrome c and diablo IAP-binding mitochondrial protein (DIABLO/Smac) are released into the cytosol, leading to mitochondrial outer membrane permeabilization (MOMP) [100,101]. Subsequently, cytochrome c binds to the apoptotic peptidase activating factor 1 (APAF1) and pro-caspase 9, to form an apoptosome complex [102]. Caspase-9, an initiator caspase, is activated by the apoptosome via heterodimerization with APAF1 and a self-homodimerization process [103,104]. This complex activates caspase-3 and caspase-7, which produce DNA fragmentation, phosphatidylserine (PS) externalization and apoptotic blebs [105,106,107,108]. In contrast, extrinsic apoptosis is mediated by transmembrane death receptors, TNF receptor superfamily members 1A (TNFR1), 10a (TRAILR1 or DR4), and 10b (TRAILR2 or DR5), and the Fas cell surface death receptor (Fas/CD95/APO1) [64]. The binding of the endogenous ligands, FAS l to the Fas receptor and TRAIL to the TRAIL receptor, recruits the adaptor proteins Fas-associated via death domain (FADD) and TNFR1 associated via death domain (TRADD), respectively, and forms an intracellular multiprotein complex known as the death-inducing signaling complex (DISC) [109]. This process enables the recruitment and activation of caspase-8 or caspase -10 through homodimerization, leading to the corresponding activation of the executioner caspase cascade, ultimately resulting in cell death [110]. 

Using the dye, TMRE, we determined the effect of TPH compounds on the mitochondrial membrane potential in BT-20 cells. This positively the charged dye is taken up by negatively charged viable mitochondria with an intact mitochondrial membrane potential, resulting in the emission of red fluorescence when exposed to the TPH compounds [111]. The vehicle control cells emitted high levels of red fluorescence (Figure 5a). However, BT-20 cells incubated with 2 and 5 μM of TPH104c or TPH104m, for 24 h, had significantly lower levels of red fluorescence, compared to the vehicle control (**, *p* < 0.01, for both concentrations of TPH104c and **, *p* < 0.01 and ***, *p* < 0.001 for 2 and 5 μM of TPH104m, respectively) (Figure 5a,b). Carbonyl cyanide m-chlorophenyl hydrazone (CCCP), an inhibitor of oxidative phosphorylation [112], was used as the positive control for this assay. BT-20 cells incubated with 200 μM of CCCP had a significantly lower level of fluorescence, compared to the vehicle control and 2 or 5 μM of TPH104c and TPH104m (Figure 5a,b; *p* < 0.0001) [113]. These results indicated that TPH104c and TPH104 caused a loss of mitochondrial membrane potential. The loss of mitochondrial membrane potential results in the release of apoptogenic factors, such as cytochrome c, from the mitochondria, which is crucial for the activation of specific caspases to initiate apoptosis [114,115,116]. Therefore, to determine whether TPH compounds release cytochrome c, which mediates apoptosis, we determined the effect of TPH104c and TPH104m (2 and 5 μM) on the level of cytochrome c using immunofluorescence. Interestingly, TPH104c and TPH104m, at 2 or 5 μM, significantly decreased the fluorescence of cytochrome c (*p* < 0.0001 and *p* < 0.001 for 2 and 5 μM of TPH104c, respectively; *p* < 0.0001 and *p* < 0.0001 for 2 and 5 μM of TPH104m, respectively (Figure 5c,d), indicating that the levels of cytochrome c were lower in BT-20 cells incubated with TPH104c and TPH104m, for 24 h, compared to the vehicle control. In contrast, BT-20 cells incubated with 1 μM of paclitaxel, known to induce apoptotic cell death, significantly enhanced cytochrome c release (**** *p* < 0.0001), compared to the vehicle control, TPH104c, or TPH104m. These results suggest that TPH104c and TPH104m did not induce cell death by increasing the levels of cytochrome c in the cytosol of BT-20 cells.

It has been reported that ROS can cause the death of cancer cells by inducing apoptosis [29,117,118,119]. Indeed, high levels of ROS activate the p53 enzyme system, which promotes late-stage apoptosis by upregulating the levels of pro-apoptotic proteins such as BAK and BAX and downregulating the levels of anti-apoptotic proteins such as BCL-2, BCL-xL, and MCL-1 [120]. In addition, ROS induce permeabilization of the outer mitochondrial membrane by opening the mitochondrial permeability transition pore (mPTP) [121,122]. This alters the electrochemical proton gradient across the mitochondria, causing the release of pro-apoptotic molecules into the cytoplasm, which are involved in apoptotic cell death [100,123]. To determine if an increase in ROS levels is the primary cause of the loss of mitochondrial membrane permeability, we determined the effect of TPH104c and TPH104m on the intracellular levels of ROS in BT-20 cells, using the dye 2′,7′-dichlorodihydrofluorescein diacetate (H_2_DCFDA). Cellular esterases hydrolyze the acetyl groups after the dye is taken up by cells, resulting in the formation of H_2_DCF [124]. Subsequently, intracellular ROS oxidize H_2_DCF to produce 2′,7′-dichlorofluorescein, which emits green fluorescence (excitation and emission wavelengths of 485 nm and 522 nm, respectively) [125]. The incubation of BT-20 cells with 2 and 5 μM of TPH104c or 2 and 5 μM of TPH104m for 24 h did not significantly alter ROS levels compared to the vehicle control (Figure 5e,f). However, as previously reported [126], BT-20 cells incubated with 2 μM of paclitaxel significantly elevated the ROS levels (**** *p* < 0.0001), compared to the vehicle control, TPH104c, or TPH104m (Figure 5). These findings suggest that at the concentrations and incubation times used in this study, TPH104c and TPH104m did not produce their anticancer efficacy by increasing ROS levels.

### 3.5. TPH104c and TPH104m-Mediated Cell Death Is Regulated by the Protein, DRP1, a Mitochondrial Marker

The loss of mitochondrial membrane potential and decrease in cytochrome c release prompted us to determine whether TPH104c and TPH104m altered the levels of mitochondrial dynamic proteins. Mitochondria undergo continuous cycles of fission and fusion to maintain mitochondrial homeostasis and quality, and balance their population and cellular function [127]. In mitochondrial fission, DRP1 is recruited from the cytoplasm to the mitochondrial outer membrane (MOM) receptor, Fis1, in a complex with the mitochondrial fission factor (MFF) [128]. This produces an incision in the mitochondrial membrane in a GTP-dependent manner, resulting in mitochondrial fission [129]. Mitochondrial fusion is mediated by the proteins mitofusin 1 (MFN1) and mitofusin 2 (MFN2), located on the MOM, and Optic atrophy 1 (OPA1) proteins, located in the inner mitochondrial membrane [130]. BT-20 cells incubated with 2 and 5 μM of TPH104c and TPH104m significantly decreased the phospho-DRP1 to DRP1 ratio (*p* < 0.0001 for 2 and 5 μM of TPH104c, respectively, and *p* < 0.0001 for 2 and 5 μM of TPH104m, respectively), compared to the vehicle-incubated cells (Figure 6a,b). Interestingly, when compared to the vehicle control, the levels of phospho-MFF to MFF were increased in BT-20 cells incubated with 2 and 5 μM of TPH104c and TPH104m, compared to the controls but were not significant. The levels of Fis1 in BT-20 cells were not significantly altered after incubation with 2 or 5 μM of TPH104c and TPH104m, compared to the vehicle-incubated cells (Figure 6a). Furthermore, BT-20 cells incubated with 2 and 5 μM of TPH104c and TPH104m, did not significantly alter the level of the mitochondrial fusion proteins, MFN1, MFN2, and OPA1, compared to the vehicle control (Figure 6a and Appendix A). We also conducted immunofluorescence experiments to determine the effect of TPH compounds on the levels of DRP1 and phospho-DRP1 in BT-20 cells. The incubation of BT-20 cells with 2 and 5 μM of TPH104c and TPH104m for 24 h significantly decreased the levels of DRP1 (*p* < 0.01 and *p* < 0.05 for 2 and 5 μM of TPH104c, respectively; Figure 6c,d). Similarly, the levels of phosphorylated DRP1 were significantly decreased in BT-20 cells after incubation with 2 (*p* < 0.01) and 5 μM (*p* < 0.05) of TPH104c and 2 (*p* < 0.05) and 5 μM (*p* < 0.001) of TPH104m, compared to vehicle-incubated cells (Figure 6e,f).

Molecular docking studies suggested that the TPH104c and TPH104m (i.e., the ligands) adopt a highly stable configuration within the binding pocket, facilitated by numerous intermolecular interactions, including hydrogen bonding with neighboring residues. Molecular dynamics simulations revealed that the aryl moiety of both molecules reside near a deeply concealed hydrophobic subpocket, enveloped by LEU51, PRO52, ILE57, ILE111, THR59 and ILE63, and this moiety progressively penetrates deeper into this subpocket over time (Appendix A). Furthermore, these compounds interact with the nearby residues, GLY37, GLY149, LYS38, SER39 and SER35, forming hydrogen bonds. Analysis conducted with the VMD program indicated that LYS38 and ASP146 have the highest hydrogen bond occupancy, underscoring their crucial role in stabilizing the ligands within the binding pocket (Appendix A). Due to the presence of a hydroxyl group, TPH104c forms a strong hydrogen bond with the carbonyl group in the main chain, specifically between residues ARG61 and PRO62, ensuring stability. In addition to hydrogen bonds and hydrophobic interactions, both TPH104c and TPH104m consistently exhibit carbon-π and donor-π interactions [131] with PRO148 and GLN34 residues throughout the simulation trajectories. These interactions are visually represented in (Figure 6g–j). Similarly, the dose-response binding kinetics of the DRP1 recombinant protein and TPH compounds were studied using Nicoya OpenSPR. These results confirmed that DRP1 binds with the TPH compounds. TPH104c and TPH104m have a direct binding interaction with the recombinant DRP1 protein, as indicated by a binding constant (KD) of 3. 57 ± 0.7 μM, 3.89 ± 1.64 μM, and 25.2 ± 3.6 μM, respectively (Figure 6k,l)

To further determine the role of DRP1 protein in the non-apoptotic cell death induced by TPH104c and TPH104m, we generated complete and partial *DRP1* knockout models, using the TNBC cell line PAC200 (a paclitaxel-resistant variant of SUM159 cells) (Figure 7a). Because TPH104c and TPH104m decreased DRP1 levels and induced non-apoptotic cell death in TNBC cell lines, we hypothesized that knocking out the *DRP1* gene in TNBC cell lines would increase TNBC cell viability. The results of this experiment supported our hypothesis, as the IC_50_ values of TPH104c and TPH104m were increased in the *DRP1* knockout (partial KO and complete KO) PAC200 cell lines. We conducted four different cell viability assays, MTT, CTB, CTG, and SRB, to confirm the above results (Appendix A). MTT assays indicated that the cytotoxic efficacy of TPH104c increased by 1.9- (*p* < 0.01) and 2.9-fold (*p* < 0.001) in partial and complete *DRP1* KO PAC200 cells, respectively (Figure 7b). Similarly, the TPH104m IC_50_ value was increased by 2.0 (*p* < 0.01) and 2.7-fold (*p* < 0.001) in the partial KO and *DRP1* KO PAC200 cells, respectively. In the CTB assay (Figure 7c), the IC_50_ values of TPH104c in partial *DRP1* KO cells were increased by 1.3-fold in partial *DRP1* KO cells and 2.1-fold (*p* < 0.05) in complete *DRP1* KO cells. There was a 0.9-fold increase in the TPH104m IC_50_ value in the -partial *DRP1* KO cells, compared to 3.3-fold (*p* < 0.05) in the partial *DRP1* KO cells treated with TPH104m. Similarly, in the CTG assay, the IC_50_ values for the TPH compounds were significantly increased in the complete *DRP1* KO PAC200 cells after incubation with TPH104c (2.7-fold, *p* < 0.01) and TPH104m (4.7-fold, *p* < 0.05), compared to the partial *DRP1* KO PAC200 cells (1.9-fold increase for TPH104c and 3.2-fold increase for TPH104m) (Figure 7d). Finally, the SRB assay (Figure 7e) results indicated that the IC_50_ values for TPH104c (3.0-fold, *p* < 0.001) and TPH104m (6.7-fold, *p* < 0.01) were significantly increased compared to the partial *DRP1* KO cells (0.6-fold increase for TPH104c and 0.8-fold increase for TPH104m) and control PAC200 cells. Thus, the increase in the viability of TNBC cells after partial and complete *DRP1* knockout suggests that TPH104c and TPH104m induce cell death by affecting the levels of the DRP1 protein. 

Furthermore, we conducted additional experiments to determine the mechanism by which TPH104c- and TPH104m-induced cell death in control wild-type and complete *DRP1* KO PAC200 cells, by performing a morphological experiment, using the Incucyte live-cell analysis system (Figure 7f). Both cell lines were incubated with 10 μM TPH104c and TPH104m for 72 h, and images were captured using an Incucyte instrument. The control (wild-type) PAC200 cells underwent death in a process similar to that of BT-20 cells: the size of the cells incubated with 10 μM TPH104c or TPH104m gradually increased, and this ultimately caused bursting (indicated by yellow arrows in Figure 7f) and detachment from the growth surface, followed by death. Interestingly, the death of the *DRP1* knockout PAC200 cells was different from that of the control PAC200 cells, as the *DRP1* KO PAC200 cells were larger in size but died without bursting, as there were no bubble-like formations in these cells. Overall, these data indicated that the decrease in DRP1 levels by TPH104c and TPH104m played a significant part in triggering a unique non-apoptotic-like cell death.

## 4. Discussion

In this study, we investigated the mechanism by which the thienopyrimidine derivates, TPH104c and TPH104m, induced the in vitro death of TNBC cells. Our data suggested that TPH104c and TPH104m induced non-apoptotic cell death in TNBC cell lines that was characterized by the absence of apoptotic morphology (i.e., no nuclear fragmentation, no cell shrinkage or apoptotic cell bodies, and the absence of rounded cells). TPH104c and TPH104m produced cell cycle arrest at the S/G2 phases and did not significantly alter the levels of ROS. TPH104c and TPH104m did not activate either initiator or executioner caspases, and cell death was not rescued by the pan-caspase inhibitor, Z-VAD-FMK. Interestingly, TPH104c and TPH104m increased mitochondrial membrane permeabilization but significantly decreased the release of total cytochrome c. Cytochrome c is an essential component of the electron transport chain, which is involved in the transfer of electrons between complexes III and IV that generates a proton gradient across the inner membrane [132,133]. Thus, the generated proton gradient is involved in ATP synthesis through the action of ATP synthase [132,133]. Therefore, a decrease in cytochrome c levels can decrease ATP production and impair mitochondrial function. Furthermore, TPH104c and TPH104m significantly decreased the levels of the mitochondrial fission protein, DRP1, and its phosphorylated form, p-DRP1. Studies have reported that inhibiting DRP1 induces apoptosis in cancer cells and may also prevent cytochrome c release, although it induces apoptosis. [134,135]. However, here, we report the non-apoptotic cell death induced by our lead compounds that bind to DRP1, result in its downregulation, as well as cytochrome c, in TNBC. Interestingly, the anticancer efficacy of our compounds was dependent on the amount of DRP1 present in TNBC cells, compared to the wild-PAC200. Indeed, the reversal of cytotoxicity was non-significant in partial *DRP1* KO PAC200 cells and significant in complete *DRP1* KO PAC200 cells. To our knowledge, the results of our study describe a novel mechanism of thienopyrimidine compound-induced non-apoptotic cell death in TNBC via decreasing Drp1 levels.

Although an increase in mitochondrial membrane potential and loss of mitochondrial membrane potential are two major events in apoptotic cell death [136], the release of the apoptogenic factor cytochrome c is required to activate the initiator caspases, followed by activation of the executioner caspases [64,98]. In this study, TPH014c and TPH104m significantly decreased cytochrome c levels, which is an apoptogenic factor released by mitochondria when cells undergo apoptotic cell death. Clearly, TPH104c and TPH104m did not induce apoptotic cell death by increasing the levels of cytochrome c. It has been reported that the loss of the mitochondrial membrane potential and subsequent mitochondrial damage are characteristics of the Fas-associated protein with death domain (FADD)-mediated necrotic death pathway; however, cytochrome c is not released [137,138]. Further investigations must be conducted to determine if Fas contributes to cell death triggered by TPH104c and TPH104m. 

In addition, our findings revealed that the incubation of TNBC cells with TPH104c and TPH104m decreased DRP1 expression, whereas there was a trend (non-significant) in the increase in MFF expression, after cells were incubated with TPH104c and TPH104m. The upregulation of MFF has been shown to facilitate the recruitment of DRP1 to the mitochondria and initiate mitochondrial fission, independent of Fis1 [128]. Our results indicate that TPH104c and TPH104m have a minimal impact on MFF-mediated DRP1 recruitment to the mitochondria. However, DRP1 is inhibited once it is recruited to the mitochondria. It is also possible that MFF is upregulated to compensate for the decrease in DRP1 levels. Studies report that DRP1 inhibition, via *DRP1* knockout, can produce in vivo tumor suppression in pancreatic cells [139] and decrease metastasis after DRP1 silencing [38]. Qian et al. reported that loss of *DRP1* in the TNBC cell line MDA-MB-231 arrested the G2/M phase of the cell cycle, produced replication stress mediated by mitochondrial hyperfusion, and led to aneuploidy [140]. Similarly, *DRP1* knockdown and mdivi-1-induced inhibition of DRP1 suppressed mitochondrial fission and decreased TNBC cell migration [141]. The depletion of DRP1 has been reported to facilitate apoptosis in human cancer cells [135,142]. However, other studies suggest that DRP1 is necessary to trigger apoptosis, and its downregulation prevents cytochrome c release and the occurrence of apoptosis [134,143,144]. Interestingly, DRP1 mediates another form of non-apoptotic cell death, necroptosis, either by activating mitochondrial phosphatase, PGAM5 [145] or through interaction with retinoblastoma [146]. TPH104c- and TPH104m-induced cell death in BT-20 cells was not rescued upon incubation with Necrostatin-1 (RIPK1 inhibitor) [147] and necrosulfonamide (MLKL inhibitor) [148] (Appendix A), which inhibit proteins required for necroptosis. Additionally, activation of DRP1 is crucial for initiating ferroptosis, another form of programmed non-apoptotic cell death, as either mitochondrial inactivation or DRP1 ablation decreases ferroptosis [149]. In accordance with this, our data indicated that the incubation of cells with Ferrostatin-1 (an inhibitor of ferroptosis [150]) did not rescue TPH104c- and TPH104m-induced cell death **(**Appendix A). Further studies need to be performed to rule out the possibility of other forms of non-apoptotic cell death induced by TPH104c and TPH104m. However, considering the significant impact of TPH104c and TPH104m on the levels of DRP1 and MFF, additional studies will be required to determine whether these compounds also affect the structure of TNBC mitochondria, disrupt energy-producing processes, such as mitochondrial respiration and glycolysis or cause damage to the mitochondria.

## 5. Conclusions

In conclusion, our study results suggest that TPH104c- and TPH104m-mediated TNBC cell death is independent of apoptosis and regulated by DRP1, and thus, it is possible that these compounds could be used to treat cancer cells that are resistant to apoptosis, although this remains to be determined. TNBC tumors are characterized by increased mitochondrial fission and levels of DRP1 compared to peritumor tissues, and this is positively correlated with a poorer prognosis in TNBC patients [40]. Thus, it is possible that targeting DRP1 may be beneficial in the treatment of TNBC [151,152]. Therefore, the thienopyrimidine derivatives TPH104c and TPH104m could be potential candidates for treating TNBC by inducing non-apoptotic, DRP1-mediated TNBC cell death. Additional experiments must be conducted to elucidate how TPH104c and TPH104m induce non-apoptotic cell death in TNBC tumors, as this will assist us in obtaining optimal lead molecules for future preclinical development.

## Figures and Tables

**Figure 1 cancers-16-02621-f001:**
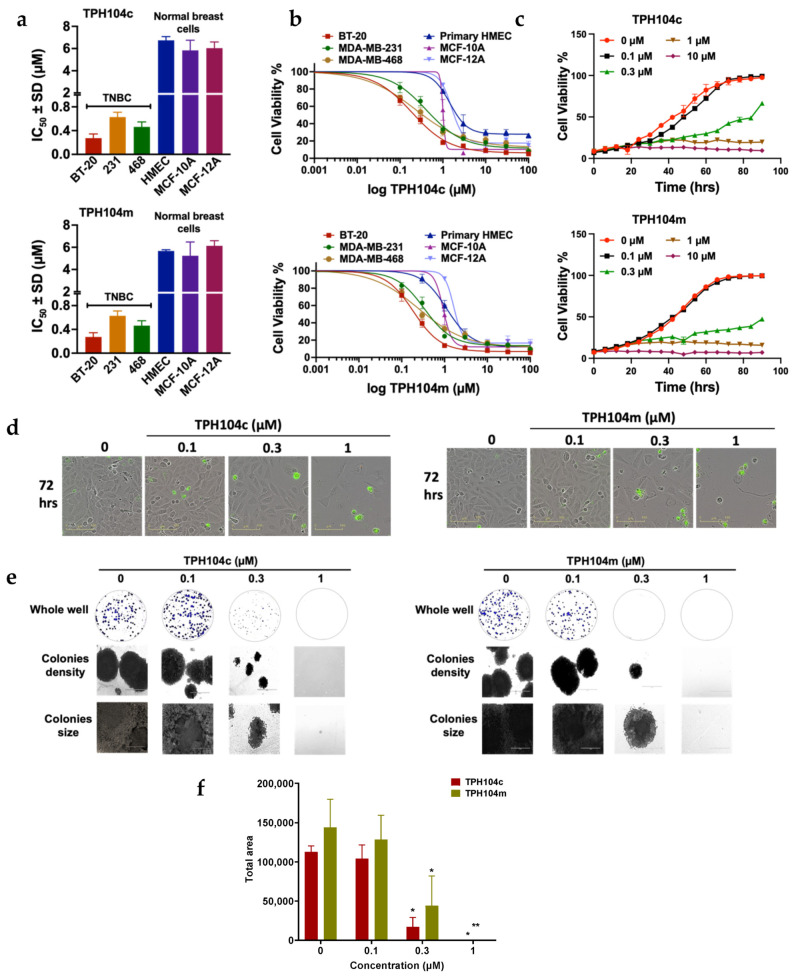
The cytotoxicity (i.e., anticancer efficacy) of TPH104c and TPH104m in different breast cancer cell lines. (**a**) The selectivity of TPH104c and TPH104m for TNBC, compared to normal, non-TNBC cell lines and TNBC, compared to normal breast cell line. (**b**) The cell viability curves of BT-20 cells after incubation for 72 h, with varying concentrations of TPH104c or TPH104m, using the MTT, CTB, or SRB assays, respectively. (**c**) Quantitative graphs of percent (%) cell viability data obtained using IncuCyte S3 software based on phase-contrast images of BT-20 cells incubated for 72 h with vehicle or varying concentrations of TPH104c, TPH104m and media. (**d**) Real-time live-cell imaging pictures of BT-20 cells after incubation with TPH104c and TPH104m for 72 hrs, in an Incucyte Cytotox green reagent—containing media. The images show the green fluorescence intensity of cytotox green dye, which stains dead or non-viable cells. (**e**) Colony formation assay for BT-20 cells that were incubated with vehicle (0 µM), 0.1, 0.3, or 1 μM of TPH104c or TPH104m. The images show the effect of TPH104c and TPH104m on colony density and size. (**f**) Bar graph summarizing the effect of different concentrations of TPH104c or TPH104m on the size of the colonies formed by BT-20 cells. The results represent the mean ± SD of three independent experiments. * *p* < 0.05, ** *p* < 0.01.

**Figure 2 cancers-16-02621-f002:**
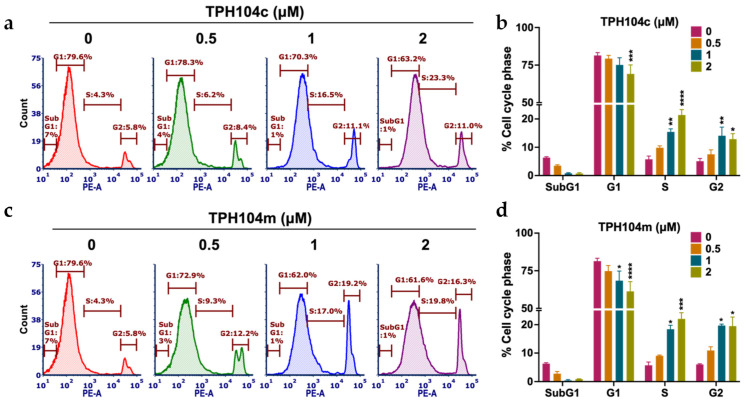
The effect of TPH104c or TPH104m on the cell cycle in BT-20 cells. Representative figures showing the distribution of BT-20 cells in different phases of the cell cycle after incubation with vehicle (0 μM), (**a**) TPH104c, or (**c**) TPH104m (0.5, 1, and 2 μM). BT-20 cells were stained with PI and subjected to flow cytometry. Count (*y*-axis) represents the cell population used in the flow cytometric analysis, and PE-A (*x*-axis) represents the cells stained with PI. Quantitative histograms depicting the percent change in BT-20 cells in the SubG1, G1, S, and G2 phases of the cell cycle upon treatment with (**b**) TPH104c or (**d**) TPH104m. * *p* < 0.05, ** *p* < 0.01, *** *p* < 0.001, **** *p* < 0.0001. The data represent the average ± SD of three separate experiments performed in triplicate.

**Figure 3 cancers-16-02621-f003:**
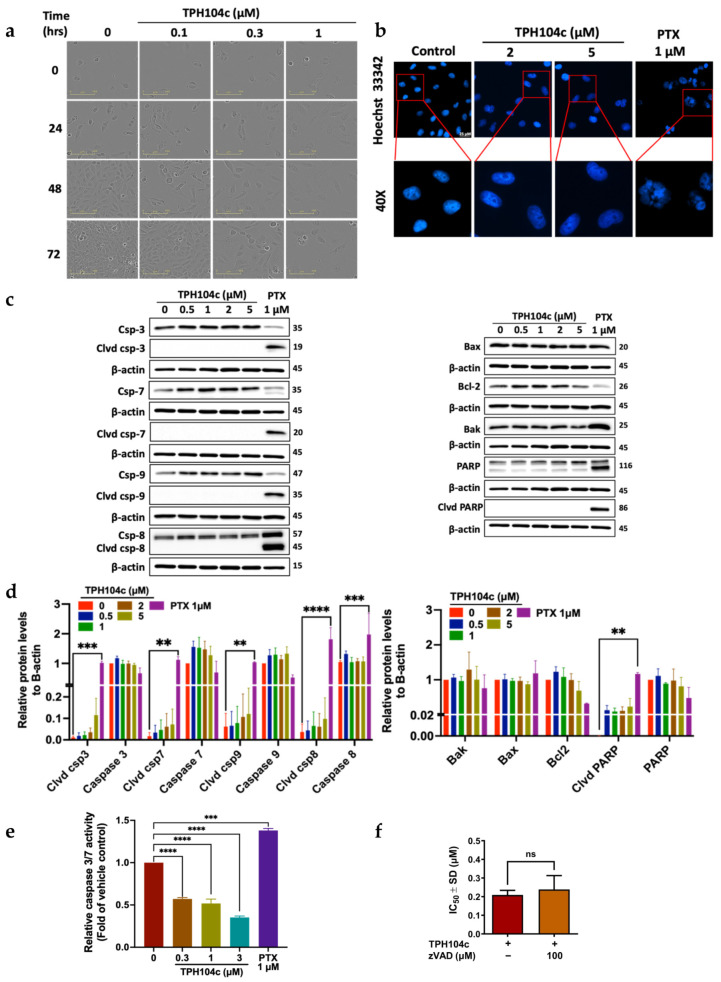
The effect of TPH104c on the levels of apoptotic and anti-apoptotic proteins in BT-20 cells. (**a**) Representative images featuring morphological changes in BT-20 cells (under 20× magnification) after incubation with vehicle (0 μM, media without drug), 0.1, 0.3, or 1 μM of TPH104c for 0, 24, 48 or 72 h. (**b**) Representative images of BT-20 cells with vehicle (0 μM), 2, or 5 μM of TPH104c for 24 h or paclitaxel (PTX, 1 μM, a positive control) and stained with Hoechst 33342 dye. TPH104c did not produce condensed or fragmented nuclei compared to cells incubated with paclitaxel (PTX). Scale bar = 25 μM. (**c**) Western blot images representing the levels of the apoptotic molecules, cleaved caspase-3, caspase-3, cleaved caspase-7, caspase-7, cleaved caspase-9, caspase-9, cleaved caspase-8, caspase-8, BAX, BAK, BCL-2, cleaved PARP and PARP, following incubation with vehicle (0 μM), 0.5, 1, 2 or 5 μM of TPH104c. The proteins are expressed as a ratio to β-actin, followed by normalization to the vehicle control. (**d**) The level of each protein is shown by histograms. Clvd = cleaved; Csp = caspase. The data represent the average ± SEM of four separate studies. (**e**) Caspase-Glo 3/7 assay results are represented as a bar graph and curve, showing a decrease in the levels of caspase-3 and caspase-7 by TPH104c, in a concentration-dependent manner in BT-20 cells, after 24 h of incubation. In contrast, 1 µMof PTX induced caspase- 3 and 7 activity (n = 2). (**f**) The IC_50_ values, using the MTT assay, for TPH104c in BT-20 cells that were preincubated with zVAD-FMK (a pan-caspase inhibitor) and then incubated with varying concentrations of TPH104c for 72 h. The data were obtained from three independent experiments conducted in triplicate and represent the average ± SD. ** *p* < 0.01, *** *p* < 0.001, **** *p* < 0.0001 and ns means non-significant. Original Western Blot images can be found in Appendix A.

**Figure 4 cancers-16-02621-f004:**
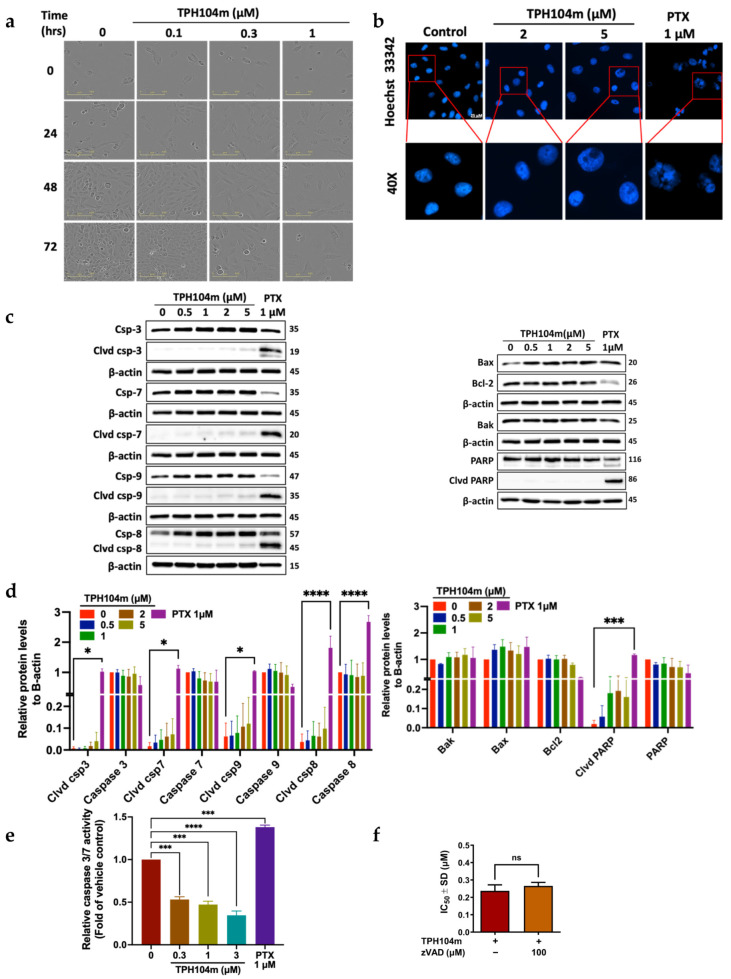
The effect of TPH104c on apoptotic and anti-apoptotic proteins in BT-20 cells. (**a**) Representative images featuring morphological changes in BT-20 cells (20× magnification) after incubation with vehicle (media without the TPH compounds or paclitaxel (PTX)), 0.1, 0.3, or 1 μM of TPH104m, at 0, 24, 48 or 72 h post-incubation. (**b**) Representative images of BT-20 cells incubated with 2 or 5 μM of TPH104m or PTX (1 μM,) a positive control) or vehicle control and stained with Hoechst 33342 dye. TPH104c did not produce condensed or fragmented nuclei, compared to cells incubated with PTX. Scale bar = 25 μM. (**c**) Western blot images for the apoptotic molecules, cleaved caspase-3, caspase-3, cleaved caspase-7, caspase-7, cleaved caspase-9, caspase-9, cleaved caspase-8, caspase-8, BAX, BAK, BCL-2, cleaved PARP, and PARP, following incubation with vehicle (0 µM), 0.5, 1, 2, or 5 μM of TPH104m. The data are expressed as the ratio to β-actin, followed by normalization to the vehicle control. (**d**) The level of each protein is shown by histograms. Clvd = cleaved; Csp = caspase. The data represent the average ± SEM of four separate studies. (**e**) Caspase-Glo 3/7 assay results are presented as a bar graph and as a curve, showing that incubation of BT-20 cells with TPH104m for 24 h decreased the levels of caspase 3/7 in a concentration-dependent manner. In contrast, PTX (1 μM) increased the levels of caspase 3 and 7 (n = 2). (**f**) IC_50_ values, using the MTT assay, for TPH104c in BT-20 cells that were preincubated with z-VADfmk and then incubated with varying concentrations of TPH104c for 72 h. The data is obtained from three independent experiments conducted in triplicates and represents the average ± SD. * *p* < 0.05, *** *p* < 0.001, **** *p* < 0.0001 and ns means non-significant. Original Western Blot images can be found in Appendix A.

**Figure 5 cancers-16-02621-f005:**
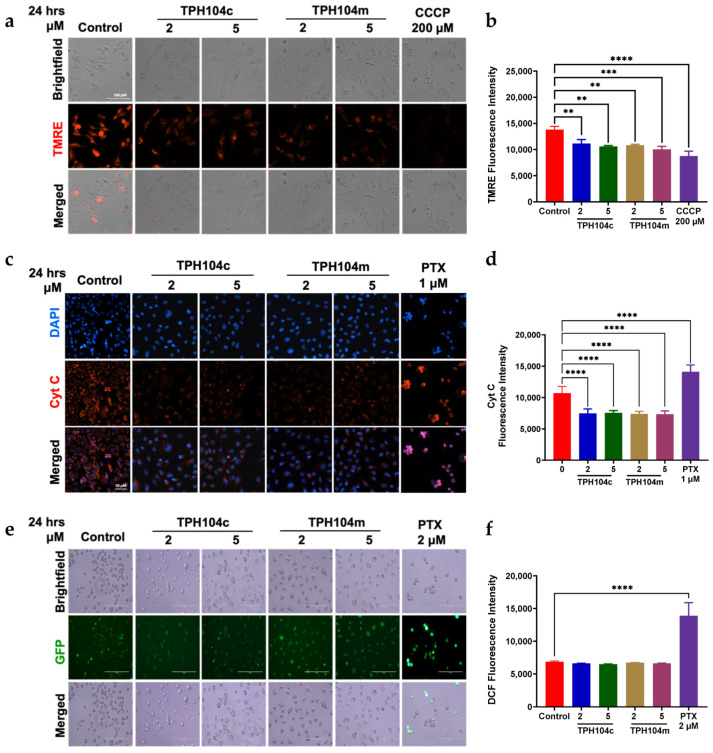
TPH104c and TPH104m induced the loss of the mitochondrial membrane potential but did not induce oxidative stress in BT-20 cells. (**a**) Fluorescent microscopic images of BT-20 cells stained with TMRE dye after incubation with the vehicle for 24 h (0 μM), 2 or 5 μM of TPH104c or TPH104m, and CCCP as a positive control. The TMRE dye is retained in cells with normal structural and functioning mitochondria, producing a high level of red fluorescence, whereas weak or no fluorescence occurred in cells without MMP. Scale bar = 200 µm. (**b**) Quantitative bar graph illustrating the change in the percentage of red fluorescence in BT-20 cells incubated with 2, or 5 µM of TPH104c and TPH104m or CCCP, compared to cells incubated with media. The results are shown as mean ± SD in triplicate. CCCP = Carbonyl cyanide 3-chlorophenylhydrazone. (**c**) Immunofluorescence analysis of cytochrome c levels in BT-20 cells incubated with 2 or 5 μM of TPH104c or TPH104m or PTX or vehicle control (0 μM), for 24 h. PTX = Paclitaxel. Scale bar = 50 µm. (**d**) Bar graphs illustrating the fluorescence intensity of cytochrome c in BT-20 cells incubated with 2 and 5 µM TPH104c and TPH104m or vehicle control (0 μM) for 24 h. (**e**) Representative images and (**f**) bar graphs depicting the level of dichlorofluorescein (DCF) fluorescence in BT-20 cells incubated with TPH104c and TPH104m (0 μM (vehicle), 2, or 5 μM) for 24 h, or paclitaxel (2 μM) for 2 h. Images were captured at 20× magnification. Scale bar = 200 µm. Relative fluorescence units of H_2_DCFA in BT-20 cells. The data are expressed as the average fluorescence ± SEM of three separate experiments. ** *p* < 0.01, *** *p* < 0.001, **** *p* < 0.0001, compared to the vehicle control cells.

**Figure 6 cancers-16-02621-f006:**
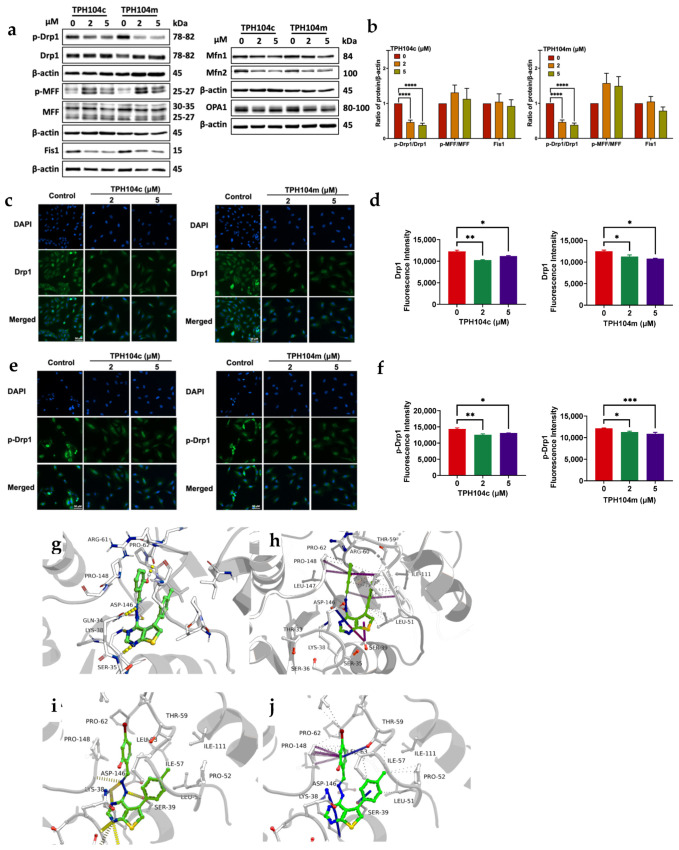
The effect of TPH104c and TPH104m on the levels of mitochondrial proteins, DRP1 and phosphorylated DRP1 (p-DRP1). (**a**) Western blot images for the mitochondrial fission proteins, p-DRP1, DRP1, p-MFF, MFF, and Fis1, and the mitochondrial fusion proteins, MFN1, MFN2, or OPA1, following incubation with vehicle (0 μM), 2, or 5 μM of TPH104c and TPH104m. All proteins were expressed as a ratio to β-actin, followed by normalization to the vehicle control. (**b**) Histograms showing the ratio of phosphorylated proteins to total proteins and individual proteins. All data are presented as the mean ± SEM of 4-5 independent studies. Immunofluorescence analysis of DRP1 (**c**) and p-DRP1 (**e**) at Serine 616C in BT-20 cells incubated for 24 h with vehicle (0 μM), 2, or 5 μM of TPH104c or TPH104m. Bar graphs showing the quantification of the fluorescence intensity of DRP1 (**d**) and p-DRP1 (**f**). Scale bar = 50 µm. * *p* < 0.05, ** *p* < 0.01, *** *p* < 0.001, **** *p* < 0.0001. Predicted non-covalent interactions of ligands TPH104c and TPH104m. (**g**) Hydrogen bonds (yellow) shared between TPH104c and DRP-1; (**h**) Carbon-π and donor-π interactions between TPH104c and DRP-1 (**i**) Hydrogen bonds (yellow) shared between TPH104m and DRP-1; (**j**) Carbon-π and donor-π interactions between TPH104m and DRP1. Representative graphs obtained from a Nicoya SPR assay, where a direct drug-protein binding interaction occurred between the Drp1 recombinant protein and varying concentrations of (**k**) TPH104c (**l**) TPH104m. Results are shown as the mean ± SD of triplicate experiments. Original Western Blot images can be found in Appendix A.

**Figure 7 cancers-16-02621-f007:**
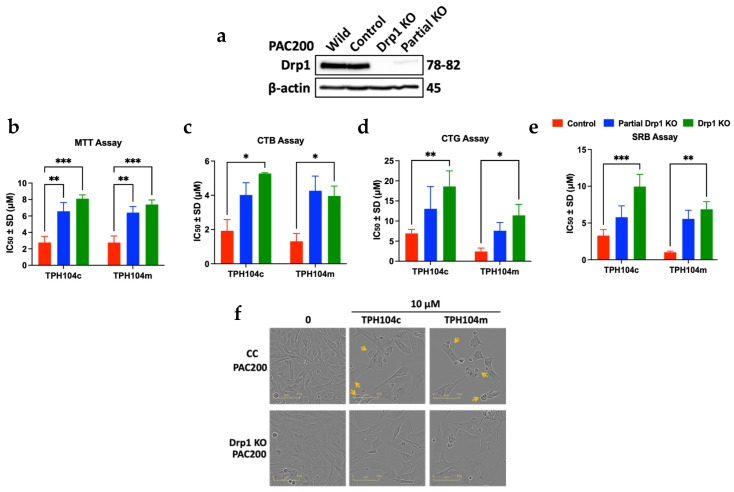
The cytotoxic efficacy of TPH104c and TPH104m on CRISPR (wild-type control) and partial and complete *DRP1* knockout (KO) PAC200 cells. (**a**) Western blot images of DRP1 levels in CRISPR wild-type (control) PAC200 cells PAC200 and complete and partial *DRP1* KO cells. Bar graphs depicting the IC_50_ values of TPH104c and TPH104m in CRISPR wild-type, partial *DRP1* KO, and complete *DRP1* KO PAC200 cells after 72 h of incubation, calculated using (**b**) MTT assay, (**c**) CTB assay, (**d**) CTG assay, and (**e**) SRB assay. (**f**) Morphological images of CRISPR wild-type and complete DRP1 KO PAC200 cells incubated with 10 μM of TPH104c and TPH104m, for 72 h. Yellow arrows represent a bubble-like formation that indicates bursting. Scale bar, 100 µm. * *p* < 0.05, ** *p* < 0.01, *** *p* < 0.001. Original Western Blot images can be found in Appendix A.

**Table 1 cancers-16-02621-t001:** The efficacy of the thieno-pyrimidin-4-yl-hydrazylidene (TPH) derivatives, TPH104c and TPH104m, in the TNBC cell line, BT-20, MDA-MB-231, and MDA-MB-468, and the normal human mammary epithelial cell lines HMEC, MCF-10A and MCF-12A.

Compounds	IC_50_ ± SD (μM)					
	TNBC			Normal		
	BT-20	MDA-MB-231	MDA-MB-468	HMEC	MCF-10A	MCF-12A
TPH104c	0.22 ± 0.06	0.48 ± 0.16	0.45 ± 0.17	6.74 ± 0.97	5.84 ± 1.81	6.04 ± 1.56
TPH104m	0.18 ± 0.03	0.47 ± 0.15	0.27 ± 0.14	5.67 ± 0.23	5.24 ± 2.47	6.13 ± 1.30

Cell survival assay was performed using the MTT assay. IC_50_ values represent the concentration required to inhibit cell proliferation by 50%. These values are presented as the average ± SD of three separate experiments conducted in triplicate. The efficacy of TPH104c and TPH104m was determined in TNBC cell lines: BT-20, MDA-MB-231, MDA-MB-468, and normal mammary epithelial cell lines: primary HMEC, MCF-10A, and MCF-12A.

**Table 2 cancers-16-02621-t002:** The efficacy of TPH104c and TPH104m in inhibiting the proliferation of BT-20 cells, as determined using the MTT, CTB and SRB assays.

Compounds	IC_50_ ± SD (μM)		
	MTT Assay	CTB Assay	SRB Assay
TPH104c	0.22 ± 0.06	0.23 ± 0.06	0.30 ± 0.07
TPH104m	0.18 ± 0.03	0.19 ± 0.08	0.32 ± 0.16
PTX	0.05 ± 0.00	0.07 ± 0.03	0.05 ± 0.00

BT-20 cell survival was further confirmed with CTB and SRB assays and compared with the MTT assay. Paclitaxel was used as the positive control. IC_50_ values represent the average concentration ± SD required to suppress cell proliferation by 50% and are the average of three separate experiments performed in triplicate.

## Data Availability

The data presented in this study are available in this article and Appendix A.

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
