# Peer review of "Novel Thienopyrimidine-Hydrazinyl Compounds Induce DRP1-Mediated Non-Apoptotic Cell Death in Triple-Negative Breast Cancer Cells"

_cancers, 2024, doi:10.3390/cancers16152621_

Round 1
Reviewer 1 Report
Comments and Suggestions for Authors
This is a very interesting paper about novel compounds which could be used in the therapy of Triple-negative breast cancer (TNBC), a tumor of extremely aggressive nature. It is an extraordinarily important problem because currently only few therapies exist against metastatic TNBC.
The compounds appear to be novel and the effects are impressive, so these findings should contribute positively to the field of cancer therapeutics, and perhaps provide the basis for further development.
Generally the manuscript is very well-written, the presented results are clear, the methodological details are in most cases sufficient, and the conclusions regarding the findings are well supported by the presented data.
However, the authors should refer to the following comments in order to improve their paper:
1. Please check the entire text for the unnecessary “space”, its hard to state it in pdf version of the paper, however in some lines there is strange break between the words (for example in lines 104, 111, 523, 905).
2. The methodology, in general, is very well-described and detailed. However in the case of cell cycle analysis determined by flow cytometer, authors should provide the missing information about the number of cells harvested and analyzed in the sample by the flow cytometer, during this experiment (in paragraph 2.6. Cell cycle analysis).
3. I was wondering, because there is no such information in the paper, what kind of sheath fluid was used during the sorting procedure? Because using original BD flow cytometer fluid leaves the cells in a very bad condition. Please include this information in the method description. (paragraph 2.14).
4. At the beginning of the results part, authors should consider again an explanation of “vehicle” term. When reading the text for the first time, it seems unclear for the reader. Its hard to find it in the paper, what the “vehicle” means in this case. Furthermore, I propose to use sometimes: “vehicle control” instead of using all the time “cells incubated with vehicle”.
5. Figures 3d and 4d are too small, it could be improved.
6. In line 579 there is a mistake – “Hoescht” instead of Hoechst.
7. There is an error in line 594 – “caspae9” instead of caspase-9.
8. Line 747 is missing a space: function[126].
9. Line 866: TPH104c- the dash is unnecessary.
10. Lines 897 and 905: unnecessary capital letters – “Our” and “The”.
11. This paper is generally well-written, however, too short discussion does not fit the amount of work and results presented in the article. Discussion clearly lacks a reference of the obtained results to the current literature. Whether there are actually no published papers on similar topic or there is really no relationships in the literature to which the obtained results could be referred?
Author Response
This is a very interesting paper about novel compounds which could be used in the therapy of Triple-negative breast cancer (TNBC), a tumor of extremely aggressive nature. It is an extraordinarily important problem because currently only few therapies exist against metastatic TNBC.
The compounds appear to be novel, and the effects are impressive, so these findings should contribute positively to the field of cancer therapeutics, and perhaps provide the basis for further development.
Generally, the manuscript is very well-written, the presented results are clear, the methodological details are in most cases sufficient, and the conclusions regarding the findings are well supported by the presented data.
Thank you for your appreciation of our work and your valuable time to review our work.
However, the authors should refer to the following comments in order to improve their paper:
- Please check the entire text for the unnecessary “space”, its hard to state it in pdf version of the paper, however in some lines there is strange break between the words (for example in lines 104, 111, 523, 905).
Thank you for drawing our attention to the issue. We have removed the unnecessary space. - The methodology, in general, is very well-described and detailed. However in the case of cell cycle analysis determined by flow cytometer, authors should provide the missing information about the number of cells harvested and analyzed in the sample by the flow cytometer, during this experiment (in paragraph 2.6. Cell cycle analysis).
We have updated the cell cycle methodology in the manuscript. A total of 100,000 cellular events were collected, with a maximum rate of 1000 events per second. - I was wondering, because there is no such information in the paper, what kind of sheath fluid was used during the sorting procedure? Because using original BD flow cytometer fluid leaves the cells in a very bad condition. Please include this information in the method description. (paragraph 2.14).
Dulbecco’s phosphate-buffered saline (DPBS) modified without Ca+2 and Mg+2 ions was utilized as a sheath fluid as recommended by BD Biosciences. - At the beginning of the results part, authors should consider again an explanation of “vehicle” term. When reading the text for the first time, it seems unclear for the reader. Its hard to find it in the paper, what the “vehicle” means in this case. Furthermore, I propose to use sometimes: “vehicle control” instead of using all the time “cells incubated with vehicle”.
We have replaced “cells incubated with vehicle” with “vehicle control” and provided a short explanation in Line 148-151 that the vehicle control are cells incubated in a drug-free me-dium containing less than 0.1% of DMSO. - Figures 3d and 4d are too small, it could be improved.
We have resized the figures. - In line 579 there is a mistake – “Hoescht” instead of Hoechst.
We fixed the spelling mistake. - There is an error in line 594 – “caspae9” instead of caspase-9.
We fixed the error. - Line 747 is missing a space: function[126].
We fixed the error. - Line 866: TPH104c- the dash is unnecessary.
We fixed the error. - Lines 897 and 905: unnecessary capital letters – “Our” and “The”.
We fixed the error. - This paper is generally well-written; however, too short discussion does not fit the amount of work and results presented in the article. Discussion clearly lacks a reference of the obtained results to the current literature. Whether there are actually no published papers on similar topic or there is really no relationships in the literature to which the obtained results could be referred?
Thank you for the question. We have updated the discussion (Line 879-897) to include current literature that has shown both the role of Drp1 in apoptosis and, on the contrary, the role of Drp1 in avoiding apoptosis. Likewise, Drp1 is also involved in other types of programmed cell death, such as necroptosis and ferroptosis. We previously investigated whether our THP104c and TPH104m induce cell death via those pathways. We have added the data to the supplementary file. Furthermore, no studies have yet demonstrated the involvement of Drp1 in this unique type of cell death, characterized by an increase in the surface area of the treated cells that resembles swelling, ultimately leading to cell death by bursting.

Reviewer 2 Report
Comments and Suggestions for Authors
In this study, the authors tested the anticancer effect of TPH104c and TPH104m in TNBC cells. Interestingly, both compounds suppressed the cell proliferation of TNBC cells compared to normal mammary epithelial cell lines. The effects were independent of apoptotic cell death. The authors have implicated that TPH104c and TPH104m cytotoxicity were associated with the downregulation of the expression of the mitochondrial fission protein, DRP1. Altogether, TPH104c and TPH104m induced non-apoptotic cell death mechanisms will play a new avenue the development of new potent and efficacious anti-cancer drugs to treat TNBC.
This is an interesting area of the study. There are a number of weakness of the studies as followinh;
1. The manuscript showing 40% similar sentence report. The author should be reduced to below 20%.
2. How thienopyrimidine derivatives compounds were reconstituted and what solvent was used to dissolve it? What vehicle is used as a control?
3. Authors should check other forms of cell death including autophagy and Ferroptosis.
4. How the quantification of proteins was done and how much proteins were loaded to determine the expression of proteins by western blots. Please mention the method.
How the DRP1 is associated with cell death in this study? Please provide evidence,
Comments on the Quality of English Languageminor English edit but 40% plagiarism
Author Response
In this study, the authors tested the anticancer effect of TPH104c and TPH104m in TNBC cells. Interestingly, both compounds suppressed the cell proliferation of TNBC cells compared to normal mammary epithelial cell lines. The effects were independent of apoptotic cell death. The authors have implicated that TPH104c and TPH104m cytotoxicity were associated with the downregulation of the expression of the mitochondrial fission protein, DRP1. Altogether, TPH104c and TPH104m induced non-apoptotic cell death mechanisms will play a new avenue the development of new potent and efficacious anti-cancer drugs to treat TNBC.
Thank you for your appreciation of our work and your valuable time to review our work.
This is an interesting area of the study. There are a number of weakness of the studies as following;
- The manuscript showing 40% similar sentence report. The author should be reduced to below 20%.
Thank you for addressing the concern. We have worked on that issue. - How thienopyrimidine derivatives compounds were reconstituted and what solvent was used to dissolve it? What vehicle is used as a control?
TPH104c and TPH104m were prepared in DMSO and then diluted in media to achieve the desired concentration. Whereas the vehicle control (cells incubated in a drug-free medium) contained less than 0.1% of DMSO. We have included this information in Line 148-151. - Authors should check other forms of cell death including autophagy and Ferroptosis.
Thank you for your suggestion. We have indeed explored other type of non-apoptotic cell death such as necroptosis and ferroptosis. We have treated the cells with inhibitors of key players of necroptosis such as Necrostatin-1 (RIPK1 inhibitor) [149], Necrosulfonamide (MLKL inhibitor) but it failed to rescue TPH104c and TPH104m-induced cell death. Similarly, ferrostatin-1, an inhibitor of ferroptosis, did not rescue TPH104c and TPH104m-induced cell death in BT-20 cells, thereby ruling out the possibility that the cell death was due to ferroptosis. Further investigations are required to rule out other non-apoptotic cell death such as autophagy, pyroptosis. We have mentioned this limitation in the discussion section. - 4. How the quantification of proteins was done and how much proteins were loaded to determine the expression of proteins by western blots. Please mention the method.
We loaded 30 mg of the proteins. Likewise, cellular proteins were quantified as the ratio to β-actin and were normalized to the vehicle control. We have provided this information in line 274 and 284-285. - How the DRP1 is associated with cell death in this study? Please provide evidence.
Thank you for your question. We have included the evidence in results section 3.5, page 23. To determine the role of the Drp1 protein in the non-apoptotic cell death induced by TPH104c and TPH104m, we generated complete and partial Drp1 knockout models using the TNBC cell line, PAC200 (a Paclitaxel-resistant variant of SUM159 cells) (Figure 7a). Because TPH104c and TPH104m decreased Drp1 levels and induced non-apoptotic cell death in the TNBC cell lines, we hypothesized that knocking out the DRP1 gene from the TNBC cell lines would increase TNBC cell viability. The results of this experiment supported our hypothesis as the IC50 values of TPH104c and TPH104m were increased in the Drp1 knockout (partial KO and complete KO) PAC200 cell lines (Figure S5). These findings were validated using 4 different cytotoxicity assays: MTT, CTB, CTG and SRB. We also performed a morphological experiment in which partial and complete Drp1 KO cells were treated with 10 μM of TPH104c and TPH104m for 72 h using the Incucyte live-cell analysis system. The control (wild-type) PAC200 cells underwent death in a process similar to BT-20 cells: the size of the cells treated with 10 μM of TPH104c or TPH104m gradually increased, and this ultimately caused bursting (indicated by yellow arrows in Figure 7e), detachment from the growth surface, followed by death. Interestingly, the death of the Drp1 knockout PAC200 cells was different than that of the control PAC200 cells, as the Drp1 KO PAC200 cells were larger in size but died without bursting, as there were no bubble-like formations in these cells. These data indicated that the decrease in Drp1 levels by TPH104c and TPH104m played a significant part in triggering a unique non-apoptotic like cell death.
Round 2
Reviewer 2 Report
Comments and Suggestions for Authors
The authors satisfactorily answered all my queries